

# Genomic signatures of globally enhanced gene duplicate accumulation in the megadiverse higher Diptera fueling intralocus sexual conflict resolution

Riyue Bao[1,2] and Markus Friedrich[3,4]

[1] Hillman Cancer Center, University of Pittsburgh, Pittsburgh, PA, USA
[2] Department of Medicine, University of Pittsburgh, Pittsburgh, PA, USA
[3] Department of Biological Sciences, Wayne State University, Detroit, MI, USA
[4] School of Medicine, Department of Anatomy and Cell Biology, Wayne State University, Detroit, MI, USA

## ABSTRACT

Gene duplication is an important source of evolutionary innovation. To explore the relative impact of gene duplication during the diversification of major insect model system lineages, we performed a comparative analysis of lineage-specific gene duplications in the fruit fly *Drosophila melanogaster* (Diptera: Brachycera), the mosquito *Anopheles gambia*e (Diptera: Culicomorpha), the red flour beetle *Tribolium castaneum* (Coleoptera), and the honeybee *Apis mellifera* (Hymenoptera). Focusing on close to 6,000 insect core gene families containing maximally six paralogs, we detected a conspicuously higher number of lineage-specific duplications in *Drosophila* (689) compared to *Anopheles* (315), *Tribolium* (386), and *Apis* (223). Based on analyses of sequence divergence, phylogenetic distribution, and gene ontology information, we present evidence that an increased background rate of gene duplicate accumulation played an exceptional role during the diversification of the higher Diptera (Brachycera), in part by providing enriched opportunities for intralocus sexual conflict resolution, which may have boosted speciation rates during the early radiation of the megadiverse brachyceran subclade Schizophora.

Corresponding author
Markus Friedrich,
ag7274@wayne.edu

## INTRODUCTION

A total of 50 years ago now, gene duplications became recognized as a major source of evolutionary innovation at the genetic level (*Ohno, 1970*). One hallmark validation of this conceptual advancement was the discovery of the Hox gene cluster, the deeply conserved string of tandem-duplicated transcription factor genes, which regulate the patterning of the longitudinal animal body axis (*Lewis, 1992*; *Carroll, 1995*; *Ryan et al., 2007*; *He et al., 2018*). Today, genome sequencing studies continue to uncover adaptive gene family expansions facilitated by gene duplication, which have affected a wide range of phenotype dimensions such as body plan innovation (*Cañestro, Yokoi & Postlethwait, 2007*; *Holland et al., 2017*; *Sakuma et al., 2019*), sensory reception (*Smadja et al., 2009*), sexual

reproduction (*Connallon & Clark, 2011*), or diet range (*Zhao et al., 2015*; *Pajic et al., 2019*), to list a few. The availability of whole genome sequences has made it also possible to quantify the global impact of gene duplication on the genetic evolution of organismal lineages. The first comprehensive study in this direction revealed that gene duplications occur at a frequency of close to 0.01 per gene per million years (*Lynch & Conery, 2000*), thus ranking equivalent with other mutational mechanisms in generating genetic variation in natural populations. The same work confirmed the expectation that most newly born gene duplicates experience rapid decay into pseudogenes. Recent studies, however, also produced evidence for a generic advantage of nascent gene duplicates in buffering gene expression noise (*Rodrigo & Fares, 2018*). Notwithstanding the nature of forces acting upon new gene duplicates, the opposing effects of gene duplication and loss events have been found to lead to rapid rates of gene turnover, which can translate into dynamic gene family size evolution(*Hahn, Han & Han, 2007*). There is, however, also a long trail of gene duplicates, which become long-term preserved due to either the complementary partitioning of ancestral pleiotropy (subfunctionalization) between sister paralogous gene duplicates or the acquisition of novel functions (neofunctionalization) by one of them (*Force et al., 1999*; *Innan & Kondrashov, 2010*). In addition, also benefits arising from genetic redundancy such as the suppression of regulatory noise can lead to hundreds of millions of years of conservation of overlapping ancestral functions in duplicated genes (*Conant & Wagner, 2004*; *Gu et al., 2003*; *Hanada et al., 2009*; *Hsiao & Vitkup, 2008*; *Tischler et al., 2006*; *Vavouri, Semple & Lehner, 2008*; *Friedrich, 2017*).

Despite these fundamental insights, much remains to be learned about the processes and mechanisms that lead to gene duplicate fixation and long-term conservation. Also the diversity, frequencies, and adaptive significance of gene duplication outcomes remain an area of continued progress in comparative genomics. In this context, it is the dichotomy of conservative or neutral vs innovative gene duplication outcomes, such as sub- vs neofunctionalization, which remains of fundamental interest in molecular evolution (*Van Hoof, 2005*; *Dean et al., 2008*; *Kondrashov, 2012*; *Qian & Zhang, 2014*; *Wang et al., 2016*; *Lan & Pritchard, 2016*; *Holland et al., 2017*; *MacKintosh & Ferrier, 2017*; *Marlétaz et al., 2018*; *Sandve, Rohlfs & Hvidsten, 2018*).

Arguably the most dramatic examples of how gene duplications provided the genetic substrate for expansions of organismal complexity and diversification are the gene duplicate enriched genomes of angiosperms in plants and of vertebrates in animals, both of which date back to several rounds of whole genome duplications (*Jiao et al., 2011*; *Cañestro et al., 2013*). A whole genome duplication has also been discovered at the base of the highly diversified arthropod subphylum Chelicerata followed by additional whole genome duplications in younger clades of this group (*Schwager et al., 2017*; *Nong et al., 2020*). At the same time, current data do not speak for an obligatory connection between gene duplication and taxonomic diversification. With close to one million documented species, insects stand at the forefront of understanding the origins of organismal diversity (*Labandeira & Sepkoski, 1993*; *Grimaldi & Engel, 2005*). However, comparative genomic evidence now solidly rules out that whole genome duplications preceded the unparalleled expansion of this group. Seeded by the *Drosophila* genome project (*Myers et al., 2000*),

the genomic exploration of insect diversity has amounted to over 100 sequenced genomes within the last 15 years (*Yin et al., 2016*). This progress is the result of targeted efforts such as i5k initiative, which strives for the completion of 5,000 high-priority arthropod genomes (*i5K Consortium, 2013*). The analysis of the first 28 genomes resulting from this effort together with 48 additional arthropod genomes suggests that gene duplicate accumulation rates remained remarkably steady over the about 450 million years of insect evolution (*Thomas et al., 2020*). The preceding analysis of the gene content in genome or transcriptome data sets of over 150 species representing all 41 insect orders, in contrast, reported evidence of substantial fluctuations in gene duplicate contents in over 15 insect orders (*Li et al., 2018*). While the strength of evidence for whole genome duplications has been refuted (*Nakatani & McLysaght, 2019*; *Roelofs et al., 2020*), these findings still leave room for the possibility that dramatically enhanced local gene duplicate retention rates played important roles in the exceptional diversification of insects.

Further noteworthy in this respect is that fact that the first reported insect genome with a dramatically higher number of lineage-specific gene duplications (close to 2,500), that of the pea aphid *Acyrthosiphon pisum* (*International Aphid Genomics Consortium, 2010*; *Julca et al., 2020*), failed to be detected in the recent study mentioned above (*Li et al., 2018*). This suggests that continued efforts are likely to refine our understanding of important gene family content changes in insect evolution. Studying the evolutionary histories of vision-related genes in insect genome models, we previously noted an unusually high number of duplicated genes in *Drosophila melanogaster* (*Bao & Friedrich, 2009*). In a followup study of over 350 developmental gene families, we discovered a preponderance of ancient, yet lineage-specific gene duplicates in *Drosophila* and the higher Diptera (Brachycera) (*Bao et al., 2018*). Moreover, more than 50% of these lineage-specific developmental gene duplications retained partial or complete genetic redundancy despite their ancient separation. This led us to hypothesize that the exceptional accumulation of developmental gene duplicates in *Drosophila* and the higher Diptera was of adaptive nature by increasing genetic robustness as a requirement for the fast development of brachyceran Diptera such as *Drosophila*. At the same time, the similarly higher number of structural vision genes gave reason to suspect the possibility of a genome-wide increase of gene duplicate accumulation in the higher Diptera (*Bao & Friedrich, 2009*; *Bao et al., 2018*).

As global studies of insect gene family contents did not produce evidence of an overall higher gene content in *Drosophila* vs non-dipteran insects (*Honeybee Genome Sequencing Consortium, 2006*; *Richards et al., 2008*; *Thomas et al., 2020*), we compared the numbers of lineage-specific gene duplicates in insect core gene families of small to moderate size (<6 paralogs). This approach was meant to eliminate the effect of adaptive gene family expansions in order to quantify the relative amounts of gene duplicate accumulation resulting from the average background rate of physical gene duplication events followed by successful fixation and longterm conservation. As a first step in this direction, we analyzed the gene contents of three distantly related, well-curated holometabolous insect genome species in comparison to *D. melanogaster*. The results from this four species comparison indicate that the megadiverse dipteran infraorder Brachycera is characterized by a genome-wide higher rate of gene duplicate accumulation.

Gene ontology analysis, however, further indicates that energy metabolism genes were exceptionally affected by this trend during the diversification of schizophoran Diptera, that is, the most recent of three major radiations in this insect order, between 40 and 60 million years ago (*Wiegmann et al., 2011*). Almost invariably, these lineage-specific energy metabolism gene duplications spawned germline-specific paralogs thereby resolving conflicting selection pressures on their ancestral singleton loci. Given the theoretical and empirical evidence that the emergence of germline-specific gene duplicates enforces species barriers, our findings point to a potentially important link between gene duplication and speciation rates in the higher Diptera in addition to documenting a higher global gene duplicate accumulation rate in this clade.

## METHODS AND MATERIALS

### Genome and sequence datasets

The genome assemblies used in this study were *Drosophila melanogaster* genome assembly 5.3 (*Myers et al., 2000*), *Mayetiola destructo*r genome assembly 1.0 (*Zhao et al., 2015*), *Anopheles gambiae* str. PEST genome database version 3.0 (*Sharakhova et al., 2007*), *Tribolium castaneum* Georgia GA2 genome database version 3.0 (*Richards et al., 2008*), and *Apis mellifera* DH4 genome database version 4.0 (*Honeybee Genome Sequencing Consortium, 2006*). The protein databases used in this study were GenBank RefSeq protein databases of *D. melanogaster*, *A. gambiae*, *T. castaneum*, and *A. mellifera* (*Pruitt, Tatusova & Maglott, 2005*) and Official gene set (OGS) protein databases of *M. destructor* (version 1.0) (*Zhao et al., 2015*), *T. castaneum* (version 1.0) (*Richards et al., 2008*), and *A. mellifera* (version 2.0) (*Honeybee Genome Sequencing Consortium, 2006*). The NCBI RefSeq and OGS protein sequence databases of *T. castaneum* and *A. mellifera* each were merged to create more comprehensive protein datasets. Four-way pairwise BLASTP searches were performed between the protein databases of *D. melanogaster*, *A. gambiae*, *T. castaneum*, and *A. mellifera*. *M. destructor* homologs were retrieved by searching *D. melanogaster* query proteins against *M. destructor* OGS protein database. In addition, *A. gambiae* genome sequences were downloaded from the GenBank RefSeq nucleotide database (*Pruitt, Tatusova & Maglott, 2005*) and searched against using *D. melanogaster* proteins as the query by TBLASTN (*Altschul et al., 1990*) to retrieve possible mosquito homologs not annotated in its RefSeq protein database. Additional genome databases interrogated in the analysis of the brachyceran enriched metabolism genes included that of the Mediterranean fruit fly *Ceratitis capitata* (*Papanicolaou et al., 2016*), the stalk-eyed fly *Teleopsis dalmanni* (*Vicoso & Bachtrog, 2015*), the calyptrate Diptera *Musca domestica* (common house fly) (*Scott et al., 2014*) and *Glossina morsitans* (tsetse fly) (*International Glossina Genome Initiative, 2014*), the onion fly *Delia antiqua* (*Zhang et al., 2014*), the robber fly species *Proctacanthus coquilletti* (*Dikow et al., 2017*), the bibionomorph species *Contarinia nasturtii* (swede midge) (GenBank assembly GCA_009176525.2) and *Sitodiplosis mosellana* (orange wheat blossom midge) (GenBank assembly GCA_009176505.1), the mosquito species *Aedes aegypti* (yellow fever mosquito) (*Nene et al., 2007*) and *Culex quinquefasciatus* (mosquito species) (*Pelletier & Leal, 2009*), and the

sandfly species *Lutzomyia longipalpis* (GCA_000265325.1) and *Phlebotomus papatasi* (GCA_000262795.1).

## Duplicate detection and classification pipeline

Gene duplication and classification were conducted by adopting the pipeline developed in our previous analysis of developmental gene duplicates (*Bao et al., 2018*). In short, species-specific protein databases (RefSeq and OGS when available) were downloaded from GenBank or respective genome project websites. For each species, databases from different sources were merged based on identical associated GeneID suggested by the Gbrowser to obtain a final protein database void of redundant sequences. Inter- and intra-species BLAST searches were performed with protein sequences of each species as queries against the database of itself or from the other species, with *E*-value cutoff set as $1.0e^{-4}$ (*Altschul et al., 1990*).

Next, gene families were sorted into three classes, (1) 1:1 orthology if the gene had 1:1 orthologs in at least two additional species, (2) ancient duplication if it resulted from a duplication that happened before the insect diversification, (3) lineage-specific duplication if this gene has independent duplication(s) occurred in any of the four species. This classification was achieved in two steps. In the first step, ortholog numbers for a given gene family of each species were determined by the following logic. Assuming there are $a$, $b$, $c$, and $d$ numbers of orthologs corresponding to 1 query gene in each of the four species, $a = b = c \geq d$ defined 1:1 orthology if each $a$, $b$, and $c$ equaled 1 and as ancient duplication if $a$, $b$, and $c$ were larger than 1. The condition $a > b = c \geq d$, by contrast, defined a family associated lineage-specific duplication in the species with more than one ortholog. The approximately 10% of gene families that did not fall into any of the three categories above were classified as unresolved groups and further analyzed in the second part of the classification analysis, which involved manual assignment of duplication labels to each gene family based on gene tree analysis results.

## Gene family validation

In order to compare the validation data from our previous analysis of a manually curated subset of 377 validation gene families (*Drăghici, 2011*; *Bao et al., 2018*) with the genome-wide duplication identification output (Supplemental Data File 2), every individual gene in the two data sets was assigned with a with a six-digit code, which reflected its inferred gene duplication history. Genes with 1:1 orthologs in all four lineages were assigned with the code 100000. Genes associated with duplication events that were shared by two or three lineages were assigned with code 010000. Code positions 3–6 indicated lineage-specific duplication events in one of the four lineages. As an example, code variant 001010 represented a gene that had independently duplicated in the lineages to *Drosophila* and *Tribolium*. In the next step, genes present in both datasets were compared directly based on their assigned duplication labels and classified into four comparison results based on each position of the six digits: (1) False positive, if the gene appeared negative in the validation dataset but positive in the genome-wide analysis; (2) False negative for genes that were positive in the validation dataset but negative in the

study; (3) True positive for genes that were positive in both results; (4) True negative for genes associated with negative codes in both datasets. Each gene was calculated redundantly in each position of the six digits, meaning that if one gene was positive for 1:1 orthology, it was counted as zero for the other five positions.

## Calculation of dS values

Protein sequences of duplicated genes were aligned with ClustalW2 (*Larkin et al., 2007*) and the corresponding cDNA alignments were generated by PAL2NAL version 13 (*Suyama, Torrents & Bork, 2006*) using the aligned protein sequences as input. Ambiguously aligned regions were removed by Gblocks 0.91b (*Talavera & Castresana, 2007*) with the block parameter "Allowed gap positions" set to "None". The gap-filtered cDNA alignments were used to calculate synonymous substitution divergence values (dS) with the yn00 algorithm of PAML version 4.4 (*Yang, 2007*). In the case of multiple duplications within the same gene family, dS values were determined for all paralog combinations and the smallest and largest values were selected for the analysis of gene duplicate age distribution.

## Phylogenetic tree reconstruction

For the genome-wide survey, ClustalW2 (*Larkin et al., 2007*) was used to generate the multiple sequence alignments, which were subsequently purged of ambiguously aligned sites and divergent regions with Gblocks (*Castresana, 2000*) applying least stringency settings, before neighbor gene gene tree reconstruction with JTT protein substitution model distances executed in Phylip package version 3.69 on Wayne State Grid supercomputing cluster (*Saitou & Nei, 1987*; *Whelan & Goldman, 2001*; *Felsenstein, 2005*). Specifically, the Seqboot module created 100 bootstrap replicates from the protein alignment input, followed by "Protdist" calculation of protein distance matrices for each bootstrap dataset applying JTT model, followed by neighbor joining tree generation for each dataset with the "Neighbor" module, and consensus tree calculation through the "Consense" module.

For the expanded analyses of gene family evolution, we collected homologs via reciprocal BLAST in the species-specific protein sequence, whole genome sequence, and TSA databases at NCBI. Gene trees were estimated with RAxML as implemented in the CIPRES Science Gateway environment from multiple protein sequence alignments generated with Clustal followed by ambiguous site removal with Gblocks at least stringent settings (*Miller, Pfeiffer & Schwartz, 2010*; *Sievers et al., 2011*; *Stamatakis, 2014*). Tree rendering and annotation was conducted in the iTOL environment (*Letunic & Bork, 2016*).

## Gene ontology analysis

The distribution of functional categories associated with lineage-specific duplication gene sets was evaluated by analysis of Gene Ontology (GO) terms (*Ashburner et al., 2000*) using the BiNGO tool (*Maere, Heymans & Kuiper, 2005*) from Cytoscape package v. 2.8 (*Smoot et al., 2011*) with customized settings: study set and reference set as listed in (Supplemental Data File 5), annotation file as the FlyBase version downloaded from Gene

Ontology official website (http://www.geneontology.org/GO.downloads.annotations.shtml, updated 08/04/2011), and ontology file as OBO version 2.0 downloaded from Gene Ontology official website (http://www.geneontology.org/GO.downloads.ontology.shtml, updated 08/04/2011), with namespace chosen as Biological Process. The statistical test was set to hypergeometric test, and multiple testing correction set to Benjamini & Hochberg False Discovery Rate (FDR) correction (*Benjamini & Hochberg, 1995*). The significance level was chosen as 0.05. GO annotations of non-*Drosophila* genes were assigned based on their orthologs in *Drosophila*. Afterwards, representative GO terms were detected and visualized by removing redundant GO terms using REViGO (*Supek et al., 2011*) with the list of significant GO terms with their associated FDR corrected *p*-values as the input, the resulting list set to small (cutoff clustering score = 0.5), the database with GO term sizes set as *Drosophila melanogaster* UniProt (*McGarvey et al., 2019*), and the semantic similarity measure method set as SimRel (*Schlicker et al., 2006*).

# RESULTS

## *Drosophila* is two-fold richer in duplicated insect core genes compared to mosquito, flour beetle, and honeybee

To probe for a genome-wide increase in gene duplicate accumulation in the lineage leading to *Drosophila*, we compared the gene family content of *D. melanogaster* with three additional insect genome model species: the mosquito *Anopheles gambiae* (Diptera: Culicomorpha), the red flour beetle *Tribolium castaneum* (Coleoptera), and the honeybee *Apis mellifera* (Hymenoptera). To normalize the data sets, we excluded species-specific orphan gene families and restricted the analysis to gene families that were conserved in at least three of the four species. To compare base rates of gene duplicate accumulation, we excluded gene families with more than six members to eliminate the effect of massively adaptive gene family expansions. In combination, these criteria narrowed the comparison to 5,983 insect core gene families conserved in *Drosophila*. Of these, 5,581, 5,505, and 5,448 were recovered in mosquito, red flour beetle, and the honeybee, respectively (Supplemental Data File 1).

Lineage-specific gene family expansions were inferred via a previously described bioinformatic pipeline involving sequence similarity threshold filtering, reciprocal BLAST tests, and, in a subset of cases, gene tree reconstruction (*Bao et al., 2018*). Running a comparison to our previously manually curated sample of 377 developmental gene families for validation analysis (*Drăghici, 2011*; *Bao et al., 2018*), we obtained evidence of 90% or better accuracy (percentage of true positives) and 85% or better specificity (percentage of true negatives) for our automated gene family analysis pipeline (Supplemental Data File 2).

In total, our bioinformatic approach detected 1,211 gene families with high confidence lineage-specific gene duplications (Supplemental Data File 1). A total of 190 of these were characterized by independent gene duplication events in more than one of the four lineages. At the species level, we found 698 gene families with lineage-specific duplications in *Drosophila*, 315 in *Anopheles*, 386 in *Tribolium*, and 223 in *Apis* (Fig. 1). Normalized for the species differences in homolog contents, 11.7% of insect core gene families were

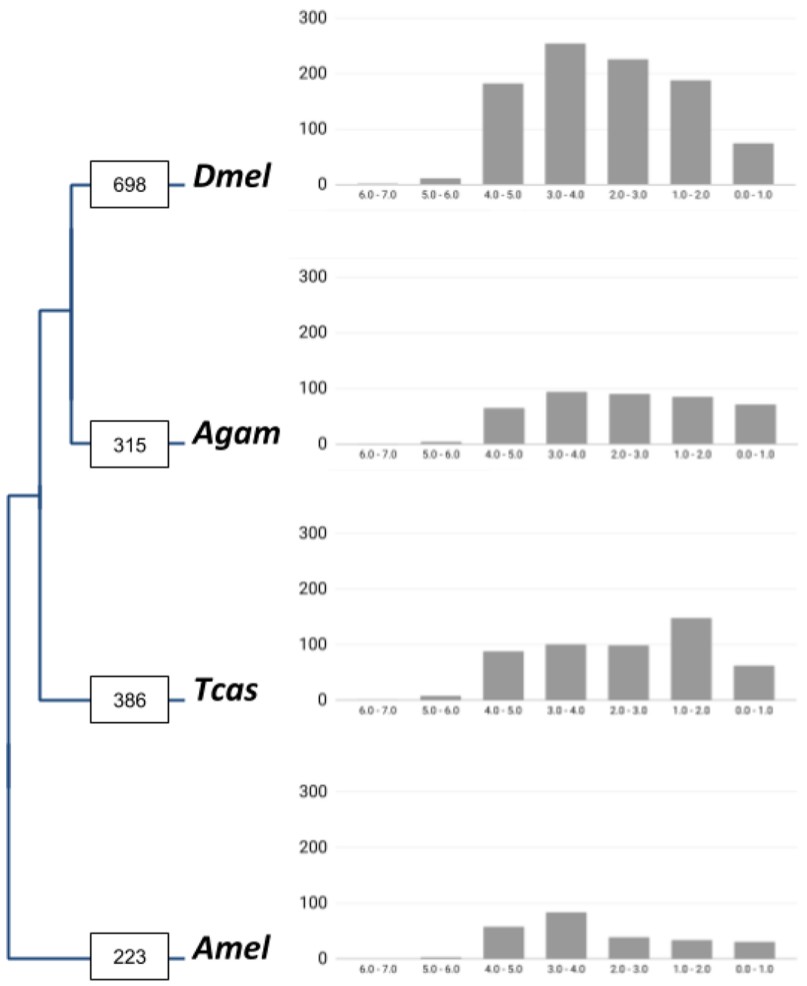

**Figure 1 Lineage-specific core gene duplicate numbers across holometabolous insect genome model species.** Boxes on terminal branches in the species tree report the numbers of gene families with lineage-specific gene duplications: Dmel, *Drosophila melanogaster*; Agam, *Anopheles gambiae*; Tcas, *Tribolium castaneum*; Amel, *Apis mellifera*. Bar charts depict distributions of lineage-specific gene duplicates by age based on neutral substitution divergence (dS) estimates with *Y*-axis representing the number of duplicates associated with each dS value bin.               

identified as expanded via lineage-specific gene duplications in the *Drosophila* lineage, which compared to 5.6%, 6.7%, and 4.1% in *Anopheles*, *Tribolium*, and *Apis*, respectively (Table 1). This on average 2-fold higher number of lineage-specific gene family expansions in *Drosophila* was consistently reflected in the large category of post-duplication 2-members large gene families and even more strongly in the gene families with 3 up to 6 extant paralogs (Table 1). Falling notably in line with the previously detected two times higher number of developmental gene duplications in the *Drosophila* lineage in the same comparative framework (*Bao et al., 2018*), these findings supported a genome-wide increase in gene duplicate accumulation as explanation for the previously detected higher numbers of both vision and developmental gene duplications in *Drosophila* vs *Anopheles*, *Tribolium*, and *Apis* (*Bao & Friedrich, 2009*; *Bao et al., 2018*).

**Table 1 Lineage-specific gene duplications in insect core gene families.** Species name abbreviations as in Fig. 1. Species-specific cell entries represent percentages of lineage-specific expanded gene families sorted by post-duplication size (2–6). Total percentages of lineage-specific expanded gene families given with absolute numbers and number of sampled gene families in parentheses.

| Gene family sizes | Dmel | Agam | Tcas | Amel | Dmel/Others |
|---|---|---|---|---|---|
| 2 | 7.9% (478) | 4.4% (246) | 4.9% (285) | 3.4% (185) | 1.9 |
| 3 | 2.4% (144) | 0.8% (42) | 1.0% (59) | 0.5% (29) | 3.1 |
| 4 | 0.9% (51) | 0.3% (19) | 0.3% (20) | 0.1% (6) | 3.2 |
| 5 | 0.3% (18) | 0.1% (5) | 0.1% (10) | 0.0001% (2) | 3.0 |
| 6 | 0.1% (7) | 0.1% (3) | 0.2% (12) | 0.02% (1) | 1.3 |
| Total | 11.7% (698/5,983) | 5.6% (315/5,581) | 6.7% (386/5,505) | 4.1% (223/5,448) | 2.1 |

## The majority of the *Drosophila*-lineage gene duplicates are old

To gain insight into the time dimensions of insect core gene duplicate accumulation in the four insect lineages, we surveyed the synonymous substitution differences (dS) between gene family paralogs as proxies of gene duplicate ages (Supplemental Data File 3). The majority of the duplicate pairs in all four lineages were associated with dS values higher than 2.0, indicative of relatively ancient origins (Fig. 1). This trend was most pronounced in the *Drosophila* lineage where 92.2% of the gene duplicate pairs were characterized by dS values higher than 2.0 compared to 82.7%, 87.9%, and 88.2% in *Anopheles*, *Tribolium*, and *Apis*, respectively.

While straightforward to compute, dS values are coarse estimates of evolutionary time dimensions, especially in the case of short gene sequences due to sample size errors. We therefore also probed for the conservation of the *Drosophila* lineage-specific gene duplications in the genome of the Hessian fly *Mayetiola destructor* (Zhao et al., 2015). This pest species is a representative of the dipteran infraorder Bibionomorpha, the now well established sister clade of the Brachycera (Wiegmann et al., 2011). Thus, duplications shared by *M. destructor* and *D. melanogaster* to the exclusion of *A. gambiae* would be diagnosed to be of pre-Brachyceran origin, while duplications unique to *D. melanogaster* to the exclusion of both *M. destructor* and *A. gambiae*, were more likely to have occurred at a later time point, that is, during the diversification of brachyceran flies.

Reciprocal BLAST searches recovered 385 (55.1%) of the 698 gene families with *Drosophila*-lineage specific duplications in *M. destructor* (Supplemental Data File 4). A total of 75 (18.5%) of these shared two or more 1:1 orthologs in the Hessian fly, implying a pre-brachyceran origin. For the remaining 185 gene families, we only detected singleton orthologs in the Hessian fly genome, characterizing them as Brachycera-specific. Thus taken together, interparalog dS divergences and gene duplicate conservation in the Hessian fly suggested that only about 20% of the close to 700 *Drosophila*-specific insect core gene duplications detected in our initial four-species comparison had accumulated before the split of the last ancestor of Hessian fly and *Drosophila*, while the majority originated later, during the expansion of the megadiverse Brachycera.

## Enrichment of energy metabolism functions in the Brachycera-specific gene duplicates

To gain insights into possible phenotypic corollaries of the heightened number of duplicated insect core genes in brachyceran Diptera, we tested for enrichment of biological processes (BP) using GO analysis tools (*Ashburner et al., 2000*; *Dennis et al., 2003*; *The Gene Ontology Consortium, 2015*). We were able to apply this approach to 1,403, 841, 860, and 426 gene duplicates of *Drosophila*, *Anopheles*, *Tribolium*, and *Apis*, respectively, based on the BP information available in FlyBase at the time of the analysis (*Drysdale & Crosby, 2005*).

Biologically meaningful GO term enrichment signals indicated informativeness of the approach (Fig. 2; Supplemental Data File 5). The GO-term "chitin metabolic processes" (GO:0006030), for instance, was exclusively enriched in the gene duplications specific to the *Tribolium*-lineage, consistent with previously reported expansions of chitin metabolism gene families in the Coleoptera (*Arakane et al., 2005*; *Dixit et al., 2008*) and the generally chitin-enriched cuticle of darkling beetles (Tenebrionidae) like *Tribolium* (*Finke, 2007*). Furthermore, the GO terms "generation of precursor metabolites and energy" (GO:0006091) and "cellular carbohydrate metabolic process" (GO:0044262) were significantly enriched in all species except *Anopheles* (Fig. 2A; Supplemental Data File 5). This genomic signal boded well with the fact that mosquitoes, in many cases, subsist on a carbohydrate-poor diet of vertebrate blood and plant pollen in contrast to the generally carbohydrate-rich diets in the clades represented by *Drosophila*, *Apis*, and *Tribolium* (*Foster, 1995*).

Interestingly, the population of *Drosophila*-lineage gene duplicates was characterized by the lowest number of enriched BP GO-terms (43) but the highest number of underrepresented BP GO-terms (206) (Fig. 2; Supplemental Data 5). Even more remarkably, the latter category included many developmental GO terms (Fig. 2; Supplemental Data 5), despite our previous finding that *Drosophila* possesses a two-fold higher number of duplicated developmental genes compared to *Anopheles*, *Tribolium*, and *Apis* (*Bao et al., 2018*). However, developmental GO terms were generally underrepresented in the lineage-specific gene duplications (Fig. 2), consistent with evidence that developmental gene regulatory networks are less tolerant to gene duplication (*Davidson & Erwin, 2006*).

In the *Drosophila* population of lineage-specific gene duplicates, significantly enriched BP GO-terms were predominantly related to energy metabolism (Fig. 2; Supplemental Data 5). Given the distinct structural and physiological features of brachyceran Diptera (*Wiegmann et al., 2011*), we also analyzed BP GO-term enrichment separately for the subsets of 234 pre-brachyceran vs 1,167 identified Brachycera-specific gene duplications. This approach detected 39 significantly enriched GO terms in the population of Brachycera-specific gene duplications all of which were related to energy metabolism. The 25 significantly enriched BP GO-terms in the population of pre-brachyceran gene duplications, in contrast, represented different categories of biological function (Fig. 2; Supplemental Data File 5).

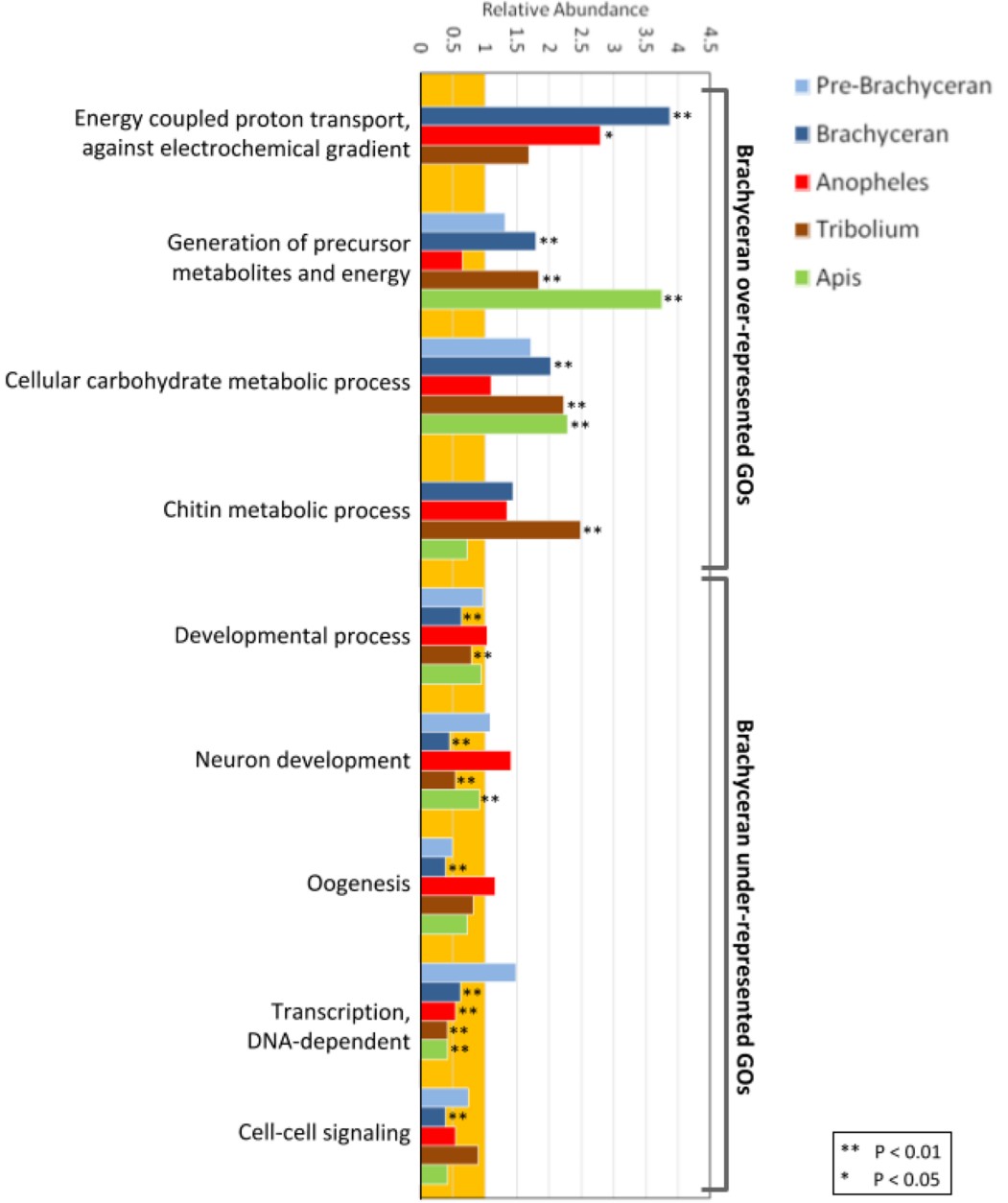

**Figure 2 Functional enrichment analysis of lineage-specific gene duplicates.** Distribution of GO functional categories in the duplicates of *Drosophila melanogaster* (Pre-Brachyceran and Brachyceran), *Anopheles gambiae*, *Tribolium castaneum*, and *Apis mellifera*. Bars represent abundance of genes associated with each listed individual GO term in the duplicates relative to those among all investigated genes (both singletons and duplicates) from each species. Relative abundance equal to one (background orange range) indicates that the GO term abundance in the duplicated genes is comparable to that in all genes, while a value lower than or higher than one indicates under- or over-representation, respectively. Brachyceran over-represented GO terms refer to the group of GO terms enriched in Brachycera-specific duplicates based on analysis of homolog conservation in *M. destructor*.

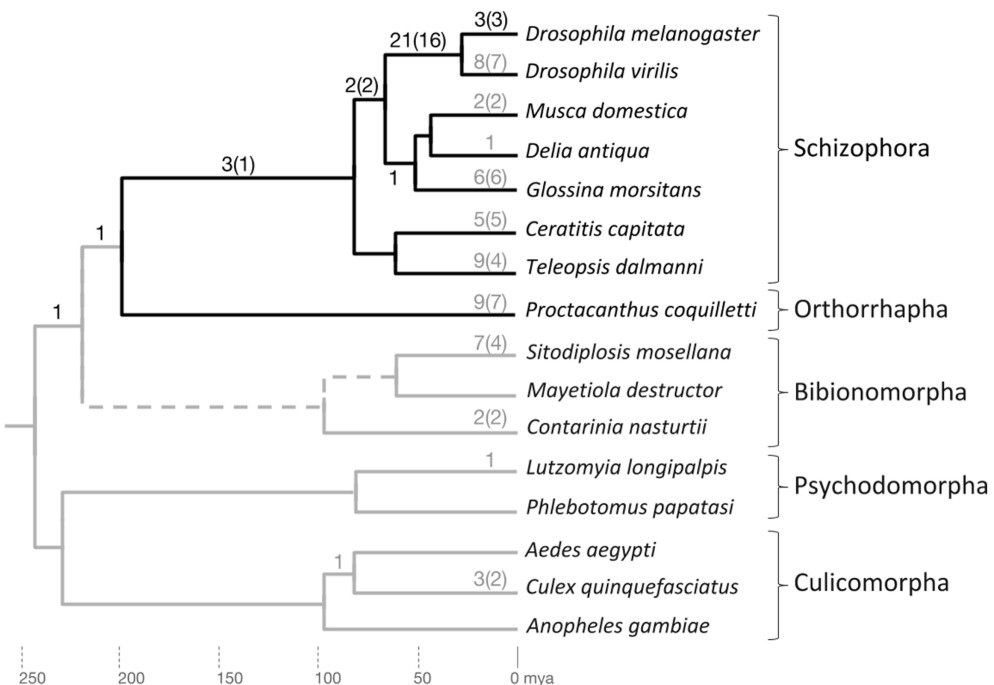

**Figure 3 Phylogenetic accumulation of brachyceran energy metabolism gene duplicates.** Summary of gene duplication time points based on homolog conservation in the species included in gene tree analysis. The Brachycera portion of the tree is indicated by dark gray branches. Dark gray numbers at branches indicate duplication events that generated conserved duplicates in the lineage to *D. melanogaster*. Light gray numbers at branches indicate parallel gene duplication events. Topology and branching time points based on *Wiegmann et al. (2011)*. mya, million years ago.

## Schizophora-concentrated origins of the *Drosophila* lineage energy metabolism-related gene duplicates

To explore the biological significance of energy metabolism-related gene duplications in brachyceran flies, we focused on gene duplicates in BP GO term categories that were significantly enriched in the Brachycera-specific gene duplications but not in any of the other investigated lineages (Supplemental Data 5). This condition was met for the BP GO terms "cell redox homeostasis" (GO:0045454), "protein targeting to mitochondrion" (GO:0006626), "protein localization in mitochondrion" (GO:0070585), "establishment of protein localization in mitochondrion" (GO:0072655), "carbohydrate phosphorylation" (GO:0046835), "dicarboxylic acid metabolic process" (GO:0043648), and "polyol metabolic process" (GO:0019751). Combined, these gene function populations constituted 189 *Drosophila* genes in 111 gene families of which, based on our initial pipeline results, 17 gene families had expanded due to Brachycera-specific gene duplications.

To scrutinize the predicted absence of these duplications outside Brachycera, we searched the genomes of additional mosquito (Suborder Culicomorpha), sandfly (Suborder Psychodomorpha), and gall midge (Suborder Bibionomorpha) species for homologs (Fig. 3). Most sampled gene families were only represented by singleton orthologs in these non-brachyceran species (Supplemental File Data 6). In two cases, however, that is, the *Thioredoxin* and *Malic enzyme* gene families, the manual homolog search raised

the initial gene family count from one to two with separate orthologs in the non-dipteran outgroups (Supplemental Data 6). Both of the *Malic enzyme* gene subfamilies contained *Drosophila*-lineage duplicates, thus increasing the number of de facto investigated gene family expansions to 18.

Only one duplication event mapped outside Brachycera to the last common ancestor (LCA) of Brachycera and Bibionomorpha (see below) (Supplemental Data File 6). Overall, the manual homolog conservation analysis confirmed the Brachycera-specificity of the energy-metabolism related gene duplications.

To explore the taxonomic depths of the metabolic gene duplications within brachyceran Diptera, we searched for homologs in a broader sample of schizophoran Diptera (Fig. 3; Supplemental Data File 6). Moreover, to differentiate between gene duplicates that dated back to the stem lineage of brachyceran Diptera vs duplicates that originated subsequently during brachyceran diversification, we searched the high coverage genome of the robber fly species *Proctacanthus coquilletti* as a representative of orthorrhaphan Brachycera (Fig. 3; Supplemental Data File 6) (*Dikow et al., 2017*).

Mapping the taxonomic distribution of gene duplicate conservation on this sampled framework of bachyceran phylogeny (*Bao et al., 2018*), we found that the majority of the energy metabolism-related gene duplications in the lineage to *Drosophila*, that is, 21 out of 32 (66%), originated in the lineage from the LCA of calyptrate + drosophilid Diptera to that of drosophilid Diptera (*D. melanogaster* and *D. virilis*) (Fig. 3). In addition, five further duplications mapped into the schizophoran clade of brachyceran Diptera: Three to the terminal branch of *D. melanogaster* representing the Sophophora subgroup of drosophilid Diptera and two back deeper to the lineage from the LCA of schizophoran Diptera to the LCA of calyptrate + drosophilid Diptera (Fig. 3).

Five duplications, finally, preceded the radiation of schizophoran Diptera (Fig. 3; Supplemental Data 6). Three of these stemmed from duplication events in a single gene family, generating the four *Thioredoxin* gene family paralogs of *D. melanogaster* that mapped to the relatively long, taxonomically still undersampled branch linking the LCAs of schizophoran and brachyceran Diptera (see below and Supplemental Data File 6). Only one duplication mapped outside Brachycera to the LCA of Brachycera and their sister clade, the Bibionomorpha (see below, Fig. 3; Supplemental Data 6). Overall, the taxonomic distribution of gene duplicate conservation was in line with the general ancientness of the Brachycera-specific gene duplications indicated by the interparalog dS divergences (Fig. 1) but also revealed the mostly schizophoran origins of the analyzed *D. melanogaster* energy metabolism gene duplicates.

## Increased duplication accumulation rate in the energy metabolism gene population during schizophoran diversification

The concentration of energy metabolism gene duplications in the schizophoran lineage to the LCA of Drosophilidae differed from the previously reported, more evenly distributed accumulation of developmental gene duplications in brachyceran Diptera. Specifically, close to 85% of the duplications in energy metabolism-related gene families mapped into the schizophoran clade of dipteran phylogeny (Fig. 3) in contrast to close to 55% of

**Table 2 Comparison of duplicate accumulation rates in developmental and energy metabolism gene families.** Rates calculated as average numbers of duplications per gene family per million years. See Fig. 4 for branch definitions. Time intervals based on *Wiegmann et al. (2011)* and *Obbard et al. (2012)*.

| Branches | Energy | Development | Time (My) |
|---|---|---|---|
| Drosophilidae LCA to present: | 0.0009 | 0.0007 | 30 |
| Drosophilidae + Calyptratae to Drosophilidae LCA: | 0.0054 | 0.0011 | 35 |
| Schizophora to Drosophilidae + Calyptratae LCA: | 0.0012 | 0.0021 | 15 |
| Brachycera to Schizophora LCA: | 0.0003 | 0.0005 | 100 |
| Brachycera + Bibionomorpha -> Brachycera LCA: | 0.0002 | 0.0004 | 50 |
| Average: | 0.0016 | 0.001 | |

developmental gene duplications (*Bao et al., 2018*). One difference between the current analysis and the previous study of developmental gene duplications was the inclusion of the robber fly *P. coquilletti* to differentiate between gene duplicates that originated in the stem lineage of brachyceran Diptera vs the lineage preceding the diversification of schizophoran species (*Bao et al., 2018*). To make the data sets more comparable, we explored the conservation of 31 duplications in 25 developmental gene families that we had previously determined to have originated prior to the diversification of schizophoran Diptera. This was accomplished by probing for evidence of homolog conservation in the robber fly *P. coquilletti*. In addition, we scrutinized for the Brachycera-specificity of these duplications by searching for homologs in the gall midge *C. nasturtii* (Supplemental Data File 7; Fig. 3). This effort mapped 18 duplications to the lineage from the LCA of brachyceran Diptera to that of Schizophora, while eight duplications mapped to the brachyceran stem lineage and three to the even older LCA of Bibionomorpha and Brachycera (Supplemental Data File 7).

Normalizing the proportions of gene duplicates per total numbers of gene families sampled in the developmental and energy-related gene populations, that is, 377 vs 111 respectively, we generated estimates of duplicate accumulation rates in the two gene populations along the brachyceran lineages leading to *Drosophila* (Table 2). Along most branches, the accumulation rates appeared similar, not exceeding 2-fold differences and averaging 0.0016 vs 0.001 duplications per gene family per million years for energy metabolism vs developmental genes, respectively. Moreover, both gene populations were characterized by a 4-fold increase in gene duplicate accumulation rate in the lineage from the LCA of schizophoran Diptera to that of Drosophilidae and calyptrate Diptera (Table 2). Most notable, however, in the schizophoran lineage connecting the LCA of Drosophilidae + Calyptratae to that of Drosophilidae the gene duplicate accumulation rate in the energy metabolism gene population peaked even further to 0.0054, exceeding that of the developmental gene population (0.0011) by a factor of 5 (Table 2). These findings suggested that duplications accumulated at higher rate during early schizophoran evolution in both gene populations except for an additional spike in the energy metabolism population in the schizophoran lineage to Drosophilidae.

**Table 3 Paralog expression specificities in the expanded energy metabolism gene families.**

| Gene families | Members | Cellular localization | Testis biased | Ovary biased | Long branch |
|---|---|---|---|---|---|
| Glutaredoxin | Grx1t | undefined | 1 | | 0 |
| | Grx1 | | digestive system | | 0 |
| Thioredoxin reductase | Trxr-1 | cytosol/mitochondrial | digestive system/testis | | 0 |
| | Trxr-2 | mitochondrial | 1 | | 1 |
| Thioredoxin | Trx-2 | nuclear | 0 | | 0 |
| | Trx-1 (dhd) | nuclear | 0 | 1 | 1 |
| | TrxT/1 | chromosome | 1 | | 1 |
| | CG13473 | – | 1 | | 1 |
| Heat Shock Protein 60 | Hsp60 | cytosol/mitochondrial | 0 | | 0 |
| | Hsp60B | | 1 | | 1 |
| | Hsp60C | | 1 | 0 | 1 |
| | Hsp60D | | 1 | | 1 |
| Mitochondrial inner membrane translocases | Tim13 | mitochondrial | 1 | | 1 |
| | CG34132 | | 0 | | 0 |
| | CG42302 | | 1 | | 1 |
| Mitochondrial inner membrane translocases | Tim17b1 | mitochondrial | 1 | | 1 |
| | Tim17b2 | | 1 | | 1 |
| | CG1724 | | 1 | | 1 |
| | Tim17b | | 0 | | 0 |
| P-P-bond-hydrolysis-driven protein transmembrane transporter 20 | Tom20 | mitochondrial | 1 | 1 | 0 |
| | tomboy20 | | 1 | 0 | 1 |
| P-P-bond-hydrolysis-driven protein transmembrane transporter 40 | Tom40 | mitochondrial | 0 | 1 | 1 |
| | tomboy40 | | 1 | 0 | 1 |
| Glycerol 3 phosphate dehydrogenase | Gpdh1 | cytosolic | muscle specific | | 0 |
| | Gpdh2 | | 1 | | 1 |
| | Gpdh3 | | 1 | | 1 |
| Glycerophosphate oxidase-1 | Gpo-1 | mitochondrial | muscle specific | | 0 |
| | Gpo-3 | | 1 | | 1 |
| | Gpo-2 | | 1 | | 1 |
| Mitochondrial anion carrier protein (MACP) gene family | Ucp4A | mitochondrial | 0 | | 0 |
| | Ucp4B | | 1 | | 1 |
| | Ucp4C | | 1 | | 1 |
| Mitochondrial carrier (TC 2.A.29) family | MME1 | mitochondrial | 1 | | 1 |
| | colt | | 0 | | 0 |
| Eukaryotic mitochondrial porin family | porin | mitochondrial | 0 | | 0 |
| | Porin2 | | 1 | | 1 |
| Hexokinase | Hex-A | cytosolic | 0 | | 0 |
| | Hex-C | | 0 | | 1 |
| | Hex-t1 | | 1 | | 1 |
| | Hex-t2 | | 1 | | 1 |

(Continued)

| Table 3 (continued) | | | | | |
|---|---|---|---|---|---|
| Gene families | Members | Cellular localization | Testis biased | Ovary biased | Long branch |
| Malate dehydrogenase 2 | Mdh2 | mitochondrial | 0 | | 0 |
| | CG10748 | | 1 | | 1 |
| | CG10749 | | 1 | | 1 |
| Succinate dehydrogenase, subunit C | SdhC | mitochondrial | 0 | | 0 |
| | CG6629 | | 1 | | 1 |
| Malic enzyme like-1 | Menl-1 | mitochondrial | 1 | | 1 |
| | Menl-2 | | 1 | | 1 |
| | Men-b | | 0 | | 0 |
| Malic enzyme | CG7848 | mitochondrial | 1 | | 1 |
| | Men | mitochondrial/cytosol | 0 | | 0 |

## Pervasive germline subfunctionalization in the enriched energy metabolism-related gene duplicate population

To gain insights into the biological significance of the energy metabolism-related gene duplications, we mined *D. melanogaster* gene expression data available through the modENCODE database (*Chen et al., 2014*). This effort revealed that all of the 18 energy metabolism-related gene families included germline-specific paralogs (Table 3). In most cases, only one paralog was documented to be expressed in a broader range of tissues. Moreover, most of the energy metabolism-related gene families (13/18) represented nuclearly encoded mitochondrial proteins (Table 3), many of which had been previously reported in testes-, or, to a much lesser extent, ovary-specific expression datasets (*Haerty et al. (2007)*: 18; *Mikhaylova, Nguyen & Nurminsky (2008)*: 6; *Wasbrough et al. (2010)*: 14; *Gallach, Chandrasekaran & Betrán (2010)*: 29).

Mapping the paralog gene expression characteristics onto gene tree topologies further revealed that germline-specificity was associated with pronounced protein sequence divergence compared to broadly expressed paralogs in the majority of cases (Supplemental Data File 8). As a paradigmatic example, the *Heat shock protein 60* (*Hsp60*) family is represented by the uniformly expressed paralog *Hsp60* and three testis-biased paralogs in *D. melanogaster*, that is, *Hsp60B*, *Hsp60C*, and *Hsp60D* (Fig. 4). Maximum likelihood gene tree estimation revealed that the uniformly expressed *D. melanogaster Hsp60* paralog is characterized by a terminal branch length that falls well within the range of that of singleton homologs of other both brachyceran and non-brachyceran species. The testis-biased paralogs, by contrast, which were unique to drosophilid species, were characterized by moderately (*Hsp60C*) to extremely extended terminal branch lengths (*Hsp60B* and *Hsp60D*) (Fig. 4).

Germline subfunctionalization combined with extremely asymmetric protein sequence evolution of sister paralog gene duplicates has been recognized to constitute a signature outcome of intralocus sexual conflict resolution (ISCR) (*Haerty et al., 2007*; *Gallach, Chandrasekaran & Betrán, 2010*). The combined evidence from gene expression and gene

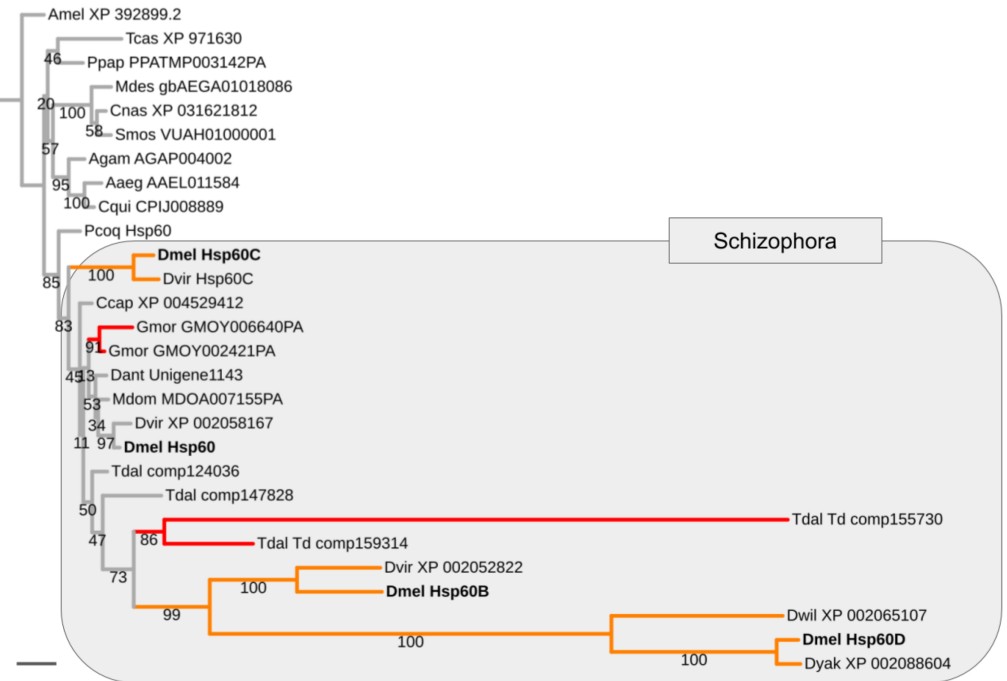

**Figure 4 Expansion and diversification of the *Hsp60* gene family in schizophoran Diptera.** Maximum likelihood tree of *Hsp60* gene family homologs compiled from dipteran species and select non-dipteran outgroup species (Amel, Tcas). Numbers at branches represent branch support from non-parametric bootstrap analysis. Branch support values lower than 50% not shown. The four *Hsp60* paralogs of *D. melanogaster* are highlighted in bold font. Orange branches indicate germline-specific paralogs. Red branches indicate parallel duplications. Species abbreviations: Aaed, *Aedes aegypti*; Agam, *Anopheles gambiae*; Ccap, *Ceratitis capitata*; Cnas, *Contarinia nasturtii*; Cqui, *Culex quinquefasciatus*; Dant, *Delia antiqua*; Dmel, *Drosophila melanogaster*; Dwil, *Drosophila willistoni*; Gmor, *Glossina morsitans*; Llon, *Lutzomyia longipalpis*; Mdes, *Mayetiola destructor*; Mdom, *Musca domestica*; Ppap, *Phlebotomus papatasi*; Pcoq, *Proctacanthus coquilletti*; Smos, *Sitodiplosis mosellana*; Tdal, *Teleopsis dalmanni*. Note: We failed to detect *Hsp60D* in *D. virilis* but found *Hsp60D* in the closely related *D. willistoni* implying that *Hsp60D* originated through a duplication that preceded the split of the Sophophora and Drosophila subgroups, which are represented by *D. melanogaster* vs *D. virilis* and *D. willistoni*, respectively. Scale bar corresponds to 0.1 substitutions per site.               

tree analyses therefore suggested that the enriched population of duplicated energy metabolism genes had been primarily deployed in ISCR events.

## Molecular signatures of intralocus sexual conflict resolution events throughout schizophoran lineages

Asking whether the duplication events in energy metabolism gene families had been of general impact in schizophoran Diptera diversity as opposed to specifically the lineage leading to *Drosophila*, we compared the numbers of independent gene duplications in the 18 investigated gene families that were detectable in the terminal branches to schizophoran and non-schizophoran species. As an example, in the *Hsp60* gene family two parallel gene duplications mapped to terminal branches of schizophoran species (*G. morsitans*, *T. dalmanni*) but no parallel duplications were detectable in the nine sampled non-schizophoran lineages (Fig. 4).

Overall, we detected 31 independent energy metabolism gene duplications in the terminal branches to the six schizophoran lineages sampled in addition to *D. melanogaster*. This compared to 9 in the robber fly *P. coquilletti* and 13 in the 8 non-brachyceran sampled dipteran species (Fig. 3). Outside Diptera, we found three parallel duplications in each *Apis* and *Tribolium* (Supplemental Data File 6). Taken together, these numbers constituted evidence in support of a Schizophora-wide increase in energy metabolism gene duplicate accumulation rate. Moreover, based on the *P. coquilleti* results, also orthorrphaphan Diptera might have experienced a relative increase in energy metabolism gene duplications. Of note, our finding of independent duplications in the *Thioredoxin*, *Hsp60*, *MME1/colt*, and *Hexokinase* gene families in the stalk eyed fly *T. dalmanni* was consistent with the results of the original study of sperm-enriched genes in *T. dalmanni* (Baker et al., 2016). Some orthology assignments differed between the two studies most likely reflecting the uncertainties coming along with low branch support values in some of the respective gene trees (Fig. 4; Supplemental Data File 8, and below).

Asking whether there was also evidence of ISCR-related trajectories in the population of independent energy metabolism gene duplication events, we tallied the fractions of parallel gene duplications that generated paralogs with pronounced asymmetric, that is, more than 2-fold, branch length differences based on maximum likelihood tree estimation results. Both parallel brachyceran gene duplications in the *Hsp60* gene family, for example, were associated with pronounced asymmetric terminal branch lengths (Fig. 4). Surveying across all energy metabolism related gene families, we found that an average of 3.0 parallel duplications with asymmetrically diverged paralogs in the schizophoran species compared to an average of 1.1 among the non-brachyceran dipteran species (Supplemental Data 6). Most of the independent gene duplications in the robber fly, however, also produced asymmetrically diverged paralogs (Fig. 3).

Comparing the average numbers of gene families in which parallel asymmetric gene duplications were detected indicated a similarly pronounced difference between the schizophoran and non-schizophoran terminal lineages (Fig. 5; Supplemental Data 6). With the caveat of lacking data on the tissue specificities for most of the of the non-*Drosophila* gene duplicates except for the stalk eyed fly *T. dalmanni* (Baker et al., 2012, 2016), these findings did amount to evidence that ISCR via gene duplication has been more common in schizophoran Diptera than in other dipteran lineages with the possible exception of orthorraphan Diptera represented by *P. coquilleti*.

## Pre-schizophoran expansion of the Drosophila Thioredoxin gene family

While our analyses indicated a the prevalence of ISCR-related duplications in the energy metabolism gene population during the diversification of schizophoran Diptera, five duplications in three energy-metabolism gene families predated this time window, based on the compilation of gene family homologs and phylogenetic gene tree analyses (Fig. 3; Supplemental Data File 6). It was therefore of interest to examine whether the protein sequence evolution characteristics of the paralogs resulting from these duplications were

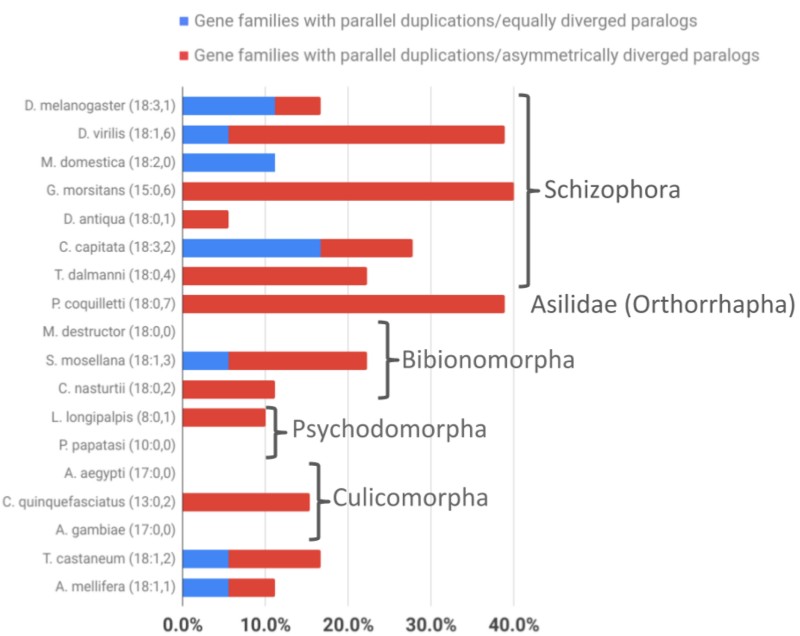

**Figure 5 Taxonomic survey of parallel duplications in the overrepresented energy metabolism gene family populations of Brachycera-specific gene duplicates.** Numbers in parentheses next to species names indicate numbers of gene families sampled followed by the number of parallel gene duplications that produced equally diverged paralogs (represented as a percentage of all gene families sampled by blue bar portions) and the number of parallel duplications with strong branch asymmetry (>2) between sister paralogs (represented as a percentage of all gene families sampled by red bar portions).

likewise indicative of ISCR trajectories despite their considerably more ancient time points of occurrence.

Three of the five pre-schizophoran gene duplications were detected in the disulfide oxidoreductase encoding *Thioredoxin* gene family, which is represented by four paralogs in *D. melanogaster*: *Trx-2* (CG31884), *Trx-1* (*deadhead*) (CG4193), *TrxT/1* (CG3315), and the yet uncharacterized locus *CG13473* (Supplemental Data File 6). While *Trx-2* is broadly expressed throughout tissues and life cycle in *D. melanogaster*, *TrxT/1* and *CG13473* transcripts are testis-enriched and *Trx-1* (*dhd*) is ovary-enriched (*Svensson et al., 2003*; *Svensson & Larsson, 2007*).

In most non-schizophoran species, including the robber fly, we only detected singleton *Thioredoxin* homologs (Fig. 6; Supplemental Data File 6). Multiple homologs were identified in the gall midge species *S. mosellana* and *C. nasturtii* as products of parallel gene duplications based on gene tree analysis results (Fig. 6). Most importantly, in the Mediterranean fruit fly *C. capitata*, we found candidate 1:1 orthologs for each of the *D. melanogaster Thioredoxin* gene family members (contig5575_1, contig5575_2, comp62603, comp61885). While branch support values were too low to draw conclusions with high confidence (number of homologous alignment sites: 75; see Supplemental Data File 7), the distribution of the *C. ceratitis Thioredoxin* homologs was best compatible with a pre-schizophoran expansion of the *D. melanogaster Thioredoxin* gene family in light of the fact that the LCA of *D. melanogaster* and *C. ceratitis* dates back to the root of

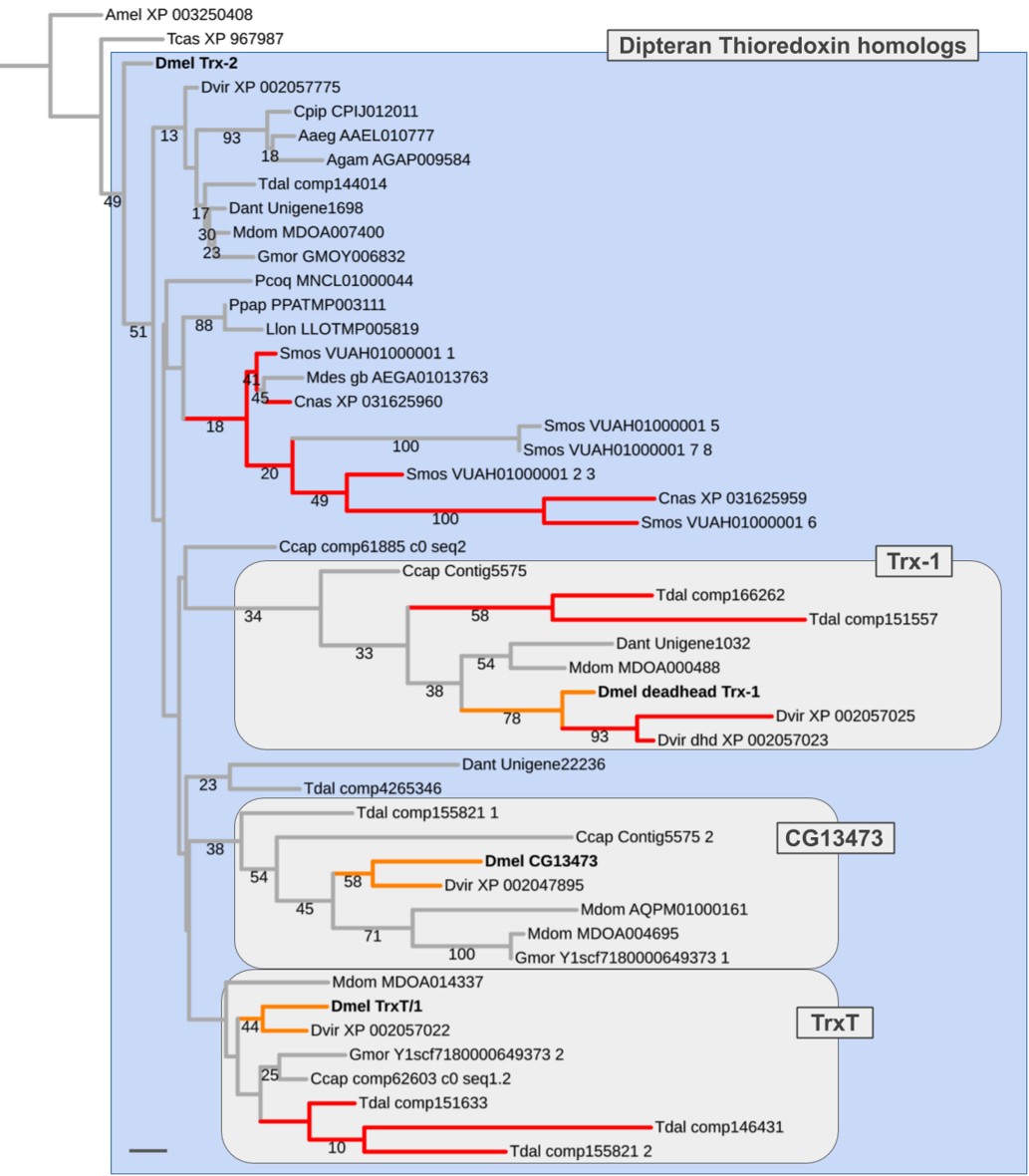

**Figure 6 Dipteran *Thioredoxin* gene family tree.** Maximum likelihood tree of *Thioredoxin* homologs compiled from dipteran species and select outgroup species. Numbers at branches represent branch support from non-parametric bootstrap analysis. Branch support values lower than 10% not shown. The four *Thioredoxin* gene family paralogs of *D. melanogaster* are highlighted in bold font. Orange branches indicate germline-specific paralogs in *Drosophila*. Red branches indicate originated asymmetric gene duplicates that originated in parallel to the duplications in the *Drosophila* lineage. Purple branches lead to robber fly homologs. Species abbreviations same as in Fig. 4. Scale bar corresponds to 0.1 substitutions per site.

schizophoran Diptera (Figs. 3 and 5) (*Wiegmann et al., 2011*). Further, consistent with the somatic, that is, likely ancestral, requirement of the *D. melanogaster* Trx-2 paralog, all non-schizophoran homologs clustered with this member of the *D. melanogaster* *Thioredoxin* gene family (Fig. 6). The germline-specific *TrxT/1*, *CG13473*, and *Trx-1* paralogs, in contrast, each clustered with a large number of bona fide orthologs from

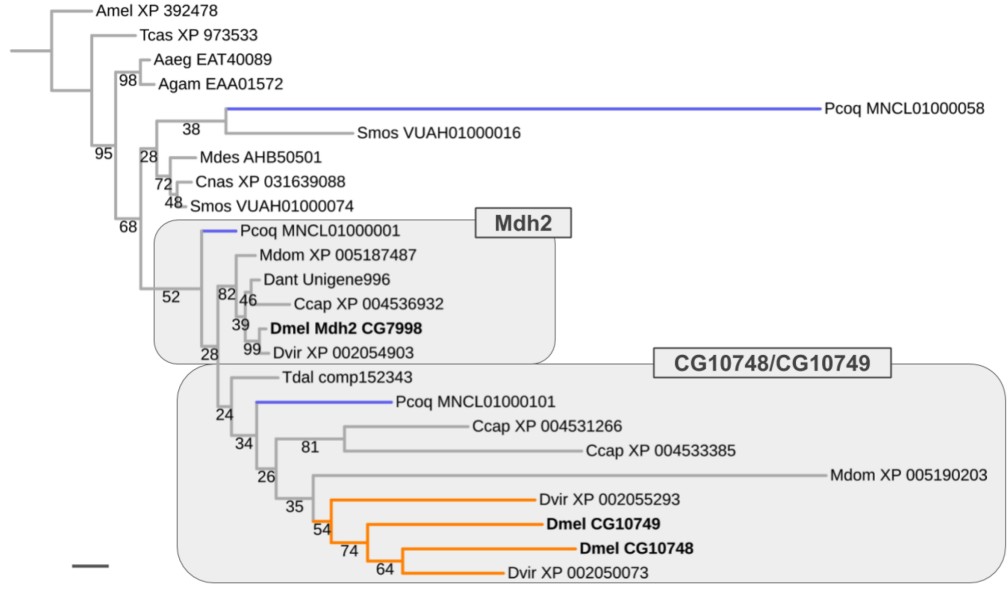

**Figure 7 Dipteran Mdh2 gene family tree.** Maximum likelihood tree of *Mdh2* homologs compiled from dipteran species and select outgroup species. Numbers at branches represent branch support from non-parametric bootstrap analysis. Branch support values lower than 50% not shown. Orange branches indicate germline-specific paralogs in the *Drosophila* lineage. Blue branches lead to robber fly homologs. Species abbreviations as in Fig. 4. Scale bar corresponds to 0.1 substitutions per site.

schizophoran species (Fig. 6). Thus overall, the gene tree analysis results suggested a pre-schizophoran ISCR-related expansion of the *Thioredoxin* gene family.

## Ancient ISCR-related expansion of the Malate dehydrogenase 2 gene family

Results similar to that obtained for the *Thioredoxin* gene family were encountered for the *Malate dehydrogenase 2* (*Mdh2*) gene family, which comprises two male germline-specific paralogs in *D. melanogaster* (CG10748, CG10749) besides *Mdh2* (CG7998) (Fig. 7). However, unlike in the case of the *Thioredoxin* gene family, evidence for the existence of not only one but three *Mdh2* gene family homologs was found in the robber fly *P. coquilletti* (Fig. 7; Supplemental Data File 6). One of them, located on genome assembly contig MNCL01000001, grouped basally with the *Mdh2* homolog cluster while the second, located on genome assembly contig MNCL01000101, branched out basally in the cluster that included the germline-specific *D. melanogaster* homologs *CG10748* and *CG10749* (Fig. 7; Supplemental Data File 6). Equivalently placed homologs were also recovered from the Mediterranean fruit fly *C. capitata* (XP 004531266, XP 004536932).

The third *P. coquilletti Mdh2* gene family homolog located on contig MNCL01000058 appeared extremely derived, branching out in the cluster of gall midge (Bibionomorpha) species homologs (Fig. 7). This was likely an artifact due to extreme amino acid substitution differences and limited number of multiple sequence alignments sites (290) for accurate gene tree reconstruction. Notwithstanding these limitations, the existence of reasonably well supported separate orthologs to the somatic and germline-specific

paralogs of the *D. melanogaster Mdh2* gene family in the robber fly *P. coquilletti* provided compelling evidence of an ISCR-related expansion of this gene family in the stem lineage to brachyceran Diptera, thus over 180 million years ago.

### Early metabolic and late germline-specific expansions in the Hexokinase gene family

The oldest duplication event in the energy metabolism gene population was detected in the Hexokinase gene family, which comprises four paralogs in *D. melanogaster*: *Hex-A* (CG3001), *Hex-C* (CG8094), *Hex-t1* (CG33102), and *Hex-t2* (CG32849) (Fig. 8; Supplemental Data File 6). While *Hex-t1* and *Hex-t2* are germline-specific paralogs, both *Hex-A* and *Hex-C* are characterized by high expression levels in a wide number of body regions based on modENCODE data and previous studies (*Moser, Johnson & Lee, 1980*; *Bourbon et al., 2002*; *Chen et al., 2014*). Moreover, *Hex-A* is expressed at higher levels than *Hex-C* in most cases except for the digestive system, head, and male testis (*Chen et al., 2014*). This correlates with an ancestral functionality of *Hex-A* in glucose metabolism in contrast to the derived affinity of *Hex-C* to fructose (*Moser, Johnson & Lee, 1980*).

Our homolog searches and gene tree analyses revealed that the broadly expressed *Hex-A* and *Hex-C* paralogs were the products of a gene duplication in the LCA of Brachycera and Bibionomorpha (Fig. 8; Supplemental Data File 6). High confidence orthologs of both *Hex-A* and *Hex-C* were detected in all dipteran species sampled with parallel duplications of *Hex-A* in the robber fly (MNCL01000216, MNCL01000057, MNCL01005250_1) and the orange wheat blossom midge *S. mosellana* (VUAH01006190, VUAH01003649) and parallel duplications of *Hex-C* in the stalk-eyed fly *T. dalmanni* (comp147884, comp160205, comp157604) (Fig. 8).

Moreover, the Hexokinase gene family tree produced strong support that the germline-specific *Hex-t1* and *Hex-t2* paralogs were born through additional duplications in the Hex-C cluster (Fig. 8). 1:1 orthologs of each germline-specific paralog, however, were only detectable in *D. virilis*, thus dating their origin prior to the diversification of drosophilid Diptera. A second cluster of independently duplicated paralogs was detected in the calyptrate species *M. domestica* and *G. morsitans* (Fig. 8). Only low support, however, was recovered for a closer relationship between these calyptrate *Hex-C* homologs and the *Drosophila Hex-t1/t2* homolog cluster.

In summary, the reconstructed sequence of gene duplication events in the Hexokinase gene family corroborated the strong association of ISCR-related gene duplication events with the schizophoran species radiation in addition to revealing an ancient duplication in the LCA of Brachycera and Bibionomorpha that most likely resulted in an expansion of metabolite usage for energy production in these clades.

## DISCUSSION

### Elevated background accumulation of insect core gene duplicates in the higher Diptera

Studying the impact of gene duplication on phenotypic evolution is of particular interest for understanding the origins of highly diverse organismal groups such as the true flies

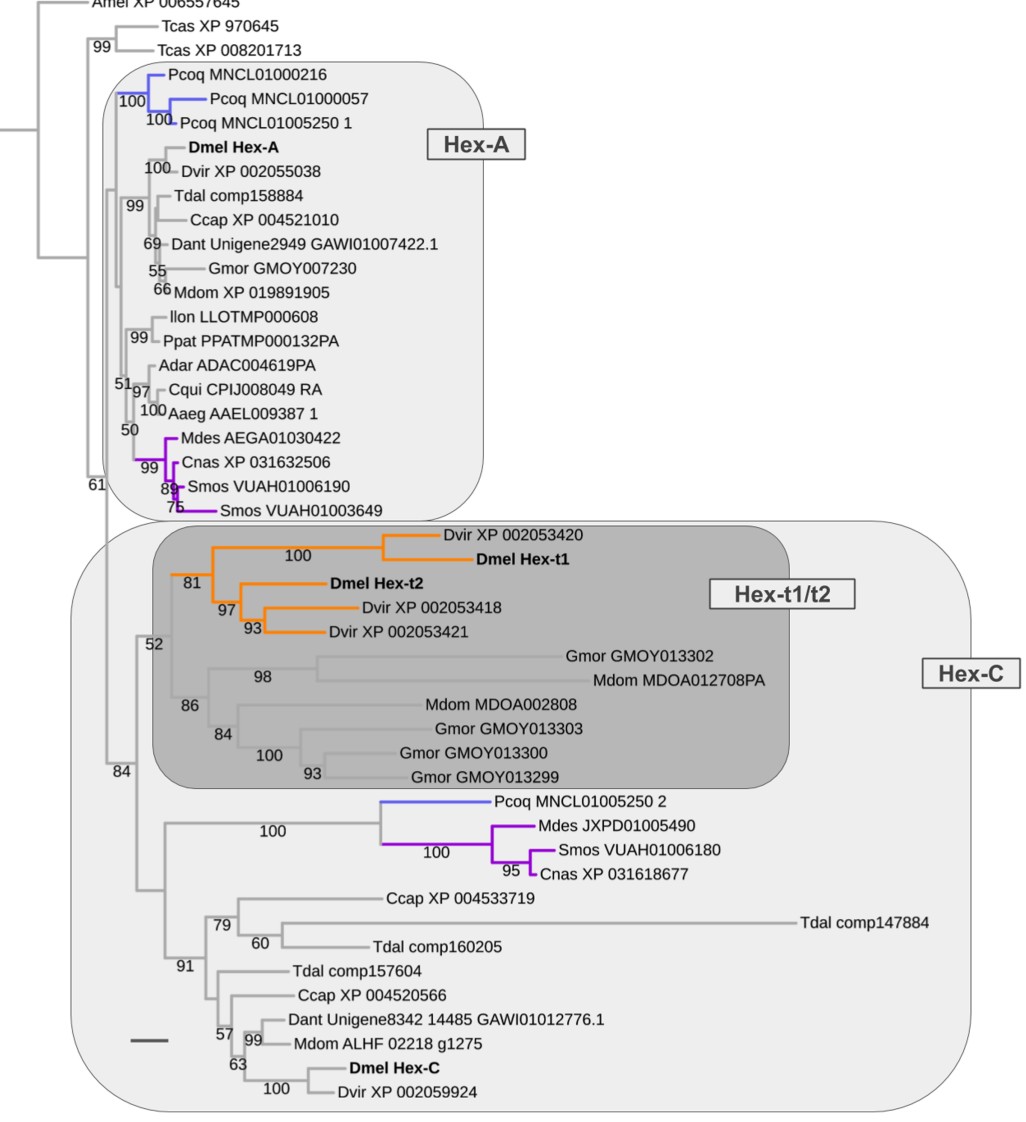

**Figure 8 Dipteran *Hexokinase* gene family tree.** Maximum likelihood tree of *Hexokinase* homologs compiled from dipteran and select outgroup species. Numbers at branches represent branch support from non-parametric bootstrap analysis. Branch support values lower than 50% not shown. Orange branches indicate germline-specific paralogs in the *Drosophila* lineage. Blue branches lead to robber fly homologs. Turquoise branches indicate gall midge (Bibionomorpha) homolog clusters. Species abbreviations same as in Fig. 4. Scale bar corresponds to 0.1 substitutions per site.

which encompass over 150,000 described species (*Yeates et al., 2007*; *Wiegmann et al., 2011*). Three major radiations have been recognized in the expansion of this megadiverse animal clade. Besides an initial rapid diversification into seven separate lineages approximately 180 million years ago, followed by the radiation of brachyceran Diptera into about 100,000 contemporary species (*Wiegmann et al., 2011*), roughly 50% of this diversity was the result of a third radiation, that of schizophoran Brachycera into over 150 families starting only about 65 million years ago (*Wiegmann et al., 2011*).

Our study was motivated by preliminary evidence that the genome of a premier representative of the Brachycera and Schizophora, i.e., *D. melanogaster*, was notably richer in lineage-specific gene duplicates compared to other equally old or older insect lineages leading to mosquito, beetle, and hymenopteran genome model species (*Bao & Friedrich, 2009*; *Bao et al., 2018*). Our findings at the genome-wide scale presented here point at an approximately 2-fold higher number of duplicated insect core gene in the lineage to *Drosophila* compared to non-brachyceran Diptera, that is, mosquitoes (Culicomorpha), honeybee (Hymenoptera), and the red flour beetle (Coleoptera). In addition to our own earlier studies, this result is consistent with multiple notions of *Drosophila*-specific gene duplications in earlier gene family studies (*Svensson, Stenberg & Larsson, 2007*; *Porcelli et al., 2007*; *Bao & Friedrich, 2009*; *Carmon & MacIntyre, 2010*; *Jiménez-Guri et al., 2013*; *Fraga et al., 2013*; *Lewis, Salmela & Obbard, 2016*; *Bao et al., 2018*; *Helleu & Levine, 2018*). At the same time, however, recent large scale analyses of gene duplication rates in insects did not detect evidence of exceptional gene gain in the *Drosophila* lineage (*Li et al., 2018*; *Thomas et al., 2020*). To the contrary, *Neafsey et al. (2015)* reported a higher gene gain rate in mosquito species compared to drosophilid Diptera. Comparing a similarly small sample of holometabolan insect lineages, *Roelofs et al. (2020)*, however, encountered evidence of a higher genome-wide gene duplication rate in the lineage to *Drosophila* in comparison to Lepidoptera, Coleoptera, and Hymenoptera.

While many high quality insect genome drafts have been generated by now, it is still reasonable to assume that the *Drosophila* genome continues to be the most comprehensively annotated and best curated one. Therefore, an alternative explanation for the larger number of detected lineage-specific gene duplicates in *Drosophila* could be lower quality of sequence coverage and gene annotation sensitivity in the genome assemblies of the non-*Drosophila* species we sampled. Three lines of evidence speak against this possible ascertainment bias: (1) The above mentioned congruence with previous gene-specific studies, (2) the high level of accuracy in our validation test, and (3) the equally high validation in our deeper analyses of gene duplications in energy metabolism gene families. Combined with our analysis of developmental gene families, the same analysis further suggests that the *Drosophila*-lineage increase of gene duplicate accumulation occurred for the most part during the early stage of the schizophoran radiation, extending from approximately 65 to 30 million years ago.

### Causation scenarios

One obvious followup question arising from our findings is which processes might have been responsible for the enhanced accumulation of duplicated insect core genes in the higher Diptera. The exclusion of adaptive gene family expansions through our focus on small size gene families and the genome-wide dimension of gene duplicate increase makes non-adaptive mechanisms appear more likely. Also the fact that the *Drosophila*-lineage gene duplicate population is characterized by the lowest number of enriched BP GO-terms (centered around a single context: energy metabolism), but the highest number of underrepresented BP GO-terms compared to the less core gene duplicate rich lineages

could be interpreted in this direction. This finding seems best explained by a gene-function independent origination mechanism that is tolerated to a similar degree by most functional contexts thus explaining the scarcity of enriched biological functions. The relatively high number of underrepresented biological functions could be envisioned to represent functional contexts that are more sensitive to the immediate consequences of gene duplications such as dosage increase. As a point in case, the underrepresentation of developmental functions in the *Drosophila* population of lineage-specific gene duplicates is consistent with the in most cases highly pleiotropic nature and dosage-sensitive action of developmental regulators, as has been noted in other cases as well (*Conant, 2020*). And yet, in the comparison between species *Drosophila* still stands out with a higher number of developmental gene duplicates (*Bao et al., 2018*), thus documenting a measurable impact of heightened gene duplicate accumulation even on this exceptionally sensitive class of genes.

Discounting adaptive forces, the higher proportion of duplicated insect core genes in the *Drosophila* lineage could be due to an increase in the rate of nonhomologous recombination, an increase in the fixation rate of nascent gene duplicates, or a decrease in gene duplicate loss rates. In principle, all of these candidate variables can be tested through comparative population genomic approaches. Sample sizes will, however, likely need to be considerable in light of the fact that our findings suggest that the 2-fold higher number of insect core gene duplicates in modern Brachycera built over a long evolutionary time span, that is, up to 65 million years, since the origin of the schizophoran stem lineage and left traces in less than 10% of the insect core gene repertoire.

In this context, it is informative to relate our findings to the more recent and dramatic expansion of the pea aphid genee content via local duplications (*International Aphid Genomics Consortium, 2010*; *Armisen et al., 2018*; *Panfilio et al., 2019*). While examples of adaptive gene family expansions have been detected for the pea aphid (*Smadja et al., 2009*), it is intriguing to note that pea aphids are characterized by cyclical parthenogenesis, that is, a mode of asexual evolution (*Miura et al., 2003*). The latter in turn implies relaxed purifying selection due to reduced effective population size, which can be hypothesized to increase the survival probability of nascent gene duplicates via genetic drift (*Lynch & Conery, 2000*). While this reference point lends support to effective population size based mechanisms as candidate explanations of the elevated gene duplicate accumulation in the higher Diptera, population genomic evidence suggests that positive selection has been the stronger effector in the fixation of gene duplicates during the more recent evolutionary history of the genus *Drosophila* (*Cardoso-Moreira et al., 2016*). Moreover, while a considerable number of parthenogenesis-capable species have been documented in flies including drosophilids (*Meyer et al., 2010*; *Gokhman & Kuznetsova, 2018*), their overall rarity gives little reason to suspect a broader impact of parthenogenesis during the early schizophoran radiation.

For these reasons, the recently identified immediate fitness benefit of gene regulatory noise suppression through gene duplications deserves equal consideration as a possible mechanism underlying the schizophoran gene duplicate accumulation increase (*Rodrigo & Fares, 2018*). Intriguingly, our study of developmental gene duplications produced

evidence for a larger amount of long-term conserved, genetically redundant paralogs in the higher Diptera compared to other insect genome models (*Bao et al., 2018*). This led us to the prediction of a higher level of genetic robustness during development, which might have benefited the comparatively fast speed of embryonic and postembryonic development in the higher Diptera. As noted above, although richer in gene duplicates compared to other insect lineages, developmental genes do not represent a significantly enriched fraction in the population of lineage-specific gene duplicates in *Drosophila*. It seems therefore reasonable to speculate that much of the functional spectrum of gene duplications in the *Drosophila* lineage produced a similar blend of conservative vs innovative functionalization outcomes as found for the developmental gene cohort. This prediction can be assessed in future studies through comprehensive analysis of expression and gene function data available for *D. melanogaster*. Even more decisive would be the integration of tissue specific expression data from a number of dipteran key species, an approach which proved highly informative in recent studies of gene duplication outcomes in vertebrates (*Marlétaz et al., 2018*).

## Possible links between enhanced gene duplicate accumulation, intralocus sexual conflict resolution, and speciation rates in the higher Diptera

Energy metabolism-related genes emerged as the only overrepresented functional category in the brachyceran gene duplicates. As previous studies noted evidence of adaptive evolution of mitochondrially encoded energy metabolism genes associated with the transition to flight in insects (*Yang et al., 2014*), we initially suspected the emergence of exceptionally fast flight capacities in the brachyceran Diptera as a possible adaptive outcome of the enriched proportion of duplicated energy metabolism genes. Our detailed analysis, however, paints a different picture. With a few notable exceptions in the *Thioredoxin* and *Mdh2* gene families, the energy metabolism gene duplications date to the early diversification of schizophoran Diptera, that is, over 100 million of years later than the primary radiation of brachyceran Diptera. Moreover, tissue-specific expression data and asymmetric paralog divergencies identified the great majority of lineage-specific energy metabolism gene duplicates as facilitators of ISCR. Consistent with this, most of our energy metabolism duplicates were previously identified as the genomic products of ISCR (*Rand, Clark & Kann, 2001*; *Gallach, Chandrasekaran & Betrán, 2010*; *Connallon & Clark, 2011*; *Wyman, Cutter & Rowe, 2012*; *Chakraborty & Fry, 2015*). In this process, the faster sequence divergence of male germline-specific paralogs is thought to be driven by higher energy requirements of competing sperm, release from conflicting functional constraints, and the higher mutagenic physiology of sperm cells due to higher radical oxygen species production (*Rettie & Dorus, 2012*; *Patel et al., 2016*; *Jiang & Assis, 2017*). Of note, ISCR by gene duplication is not considered subfunctionalization in the strict sense of leading to an adaptively neutral breakup of ancestrally pleiotropic functionality (*Gallach & Betrán, 2011*). However, it seems commonly assumed, and is testable, that the precursor homologs of ISCR paralogs have been homogeneously expressed in germline and somatic cells, thus defining the

differential expression of ISCR paralogs in germline vs somatic tissues as subfunctions of the ancestral expression repertoire.

Overall, our finding of a large number of germline-biased paralogs in the germline-biased energy metabolism gene population is of little surprise in light of the fact that over 10% of the *Drosophila* coding genome is characterized by germline biased gene expression (*Chintapalli, Wang & Dow, 2007*). More remarkable may be our finding that the outcome of ISCR by gene duplication can persist over long evolutionary time spans. The majority of the ISCR-related gene duplications in the lineage to *Drosophila* mapped to the early radiation of schizophoran Diptera, about 40–60 million years ago. Moreover, the oldest ISCR-related gene duplications captured in our analysis occurred most likely at least 80 million years ago (*Mdh2* and *Thioredoxin* gene families) (Supplemental Data File 6). For comparison, the most thoroughly studied case of gene duplication facilitated ISCR studied in *Drosophila* thus far dates back about 200,000 years (*VanKuren & Long, 2018*).

Our genomic detection of ISCR associated gene duplicates in subclades of the *Drosophila* family, that is, the *Drosophila* and *Sophophora* groups, is consistent with the notion that ISCR has continued to be of relevance during the more recent diversification of drosophilid Diptera (*Mikhaylova, Nguyen & Nurminsky, 2008*; *Kondo et al., 2017*). In addition, we detected a considerable number of parallel gene duplications with ISCR signatures, that is, extremely asymmetrically diverged sister paralogs, in other schizophoran lineages. Taken together, these findings and that of others in tephritid Diptera indicate that ISCR has been of widespread occurrence and significance in schizophoran Diptera (*Baker et al., 2016*).

Two lines of evidence indicate a possible link between ISCR frequency and speciation rate in our findings. For one, our data suggest a higher frequency of ISCR in the overall largely expanded schiziphoran Diptera compared to other dipteran clades with exception of the robber fly lineage. Second, we detect evidence of a spike of ISCR related gene duplication events during early schizophoran radiation. Together, tese findings speak to the still open debate of whether ISCR impacts reproductive isolation and therefore ultimately speciation rates (*Coyne & Orr, 1989*; *Parker & Partridge, 1998*; *Gavrilets, 2014*). Experimental studies produced mixed signals. Consistent with our results for the higher Diptera, *Katzourakis et al. (2001)* found evidence for a correlation between sexual selection and species richness in hoverflies. Similar conclusions have been drawn in a study of tephritid species (*Congrains et al., 2018*), which aligns taxonomically more closely with our findings of parallel sex biased gene duplications in schizophoran lineages outside the Drosophilidae. Based on artificial breeding experiments, however, sexual conflict was concluded to play no role in reproductive isolation in *D. pseudoobscura* (*Bacigalupe et al., 2007*). At the same time, there is evidence for the action of sexual conflict in allopatric experimental populations of *D. melanogaster* (*Syed et al., 2017*). At this point, the available evidence may still be summed up to suggest that speciation rate increase is driven by sexual conflict in some but not all clades (*Gavrilets, 2014*).

In support of the latter notion, we not only detected substantially lower numbers of parallel candidate ISCR gene duplications in non-brachyceran Diptera but also in

*Tribolium* and the honeybee, representatives of two megadiverse insect groups. Given the relatively small number and biased selection of gene families sampled, future studies will be needed to scrutinize the preliminary evidence that ISCR may have been of exceptional importance during the radiation of schizophoran Diptera compared to other megadiverse insect groups such as the Hymenoptera and Coleoptera. At this point, however, it is tempting to speculate that a heightened gene duplication background rate fueled ISCR events during the massive radiation of schizophoran Diptera. Encouragingly, our finding that the ISCR promoted germline-specificity of gene duplicates can remain stable over long evolutionary time scales is also of practical significance as it opens up venues for studying the global significance of ISCR in species diversification via comparative genomics and transcriptomics. This promises more definitive answers to the questions raised above and may potentially deliver even generally deeper insights into the role of molecular germ line subfunctionalization in animal speciation (*White-Cooper & Bausek, 2010*).

## CONCLUSIONS

Our comparative analysis of lineage-specific gene duplicates in four holometabolous insect genome species produced evidence of an enhanced gene duplicate accumulation rate in the lineage to *Drosophila*. Our phylogenetic surveys of developmental and energy metabolism gene duplicates suggest that this increase occurred largely during the diversification of schizophoran Diptera about 60 million years ago. Energy metabolism gene duplicates seem to have experienced an exceptional increase facilitating ISCR via gene duplication, which may also have impacted speciation rates during the early phase of the dramatic schizophoran radiation. Our study further shows that ISCR originated gene duplicates can remain conserved over considerable evolutionary time scales, which should facilitate broader genomic and transcriptomic studies of the relationship between ISCR and speciation rates.

## ACKNOWLEDGEMENTS

We thank the anonymous reviewers for critical input, Sorin Draghici, Chuanzhou Fan, and Jason Caravas for comments and advice on the project, Wayne State University Scientific Computing Program for providing and maintaining the Grid supercomputing cluster service.

### Funding

Riyue Bao was supported by Thomas C. Competitive Rumble University Graduate Fellowship and College of Liberal Arts and Sciences Enhancement GRA Scholarship. This project was supported by NSF award EF-0334948. The funders had no role in study design, data collection and analysis, decision to publish, or preparation of the manuscript.

## Grant Disclosures

The following grant information was disclosed by the authors:
Thomas C. Competitive Rumble University Graduate Fellowship.
College of Liberal Arts and Sciences Enhancement GRA Scholarship.
NSF: EF-0334948.

## Competing Interests

The authors declare that they have no competing interests.

## Author Contributions

- Riyue Bao conceived and designed the experiments, performed the experiments, analyzed the data, prepared figures and/or tables, authored or reviewed drafts of the paper, and approved the final draft.
- Markus Friedrich conceived and designed the experiments, performed the experiments, analyzed the data, prepared figures and/or tables, authored or reviewed drafts of the paper, and approved the final draft.

## Data Availability

This study relies on public domain genomic and transcriptomic data sources available at the RefSeq, Official Gene Set (OGS), whole genome shotgun contigs (wgs), and Transcriptome Shotgun Assemblies (TSA) databases maintained by the National Center of Biotechnology Information (NCBI). All analyzed sequences are also available in the Supplemental File.

- RefSeq: XP_001120807, XP_001121522, XP_001123018, XP_001653595, XP_001664283, XP_001687881, XP_001841997, XP_001845276, XP_001850913, XP_001850913, XP_001859594, XP_001862425, XP_002047597, XP_002047895, XP_002047939, XP_002048011, XP_002048624, XP_002049040, XP_002049720, XP_002050073, XP_002050587, XP_002050844, XP_002050845, XP_002050869, XP_002051123, XP_002051268, XP_002051368, XP_002051379, XP_002051380, XP_002051381, XP_002051422, XP_002051833, XP_002051868, XP_002052164, XP_002052597, XP_002052598, XP_002052822, XP_002053418, XP_002053420, XP_002053421, XP_002053422, XP_002053434, XP_002053611, XP_002054324, XP_002054373, XP_002054903, XP_002055038, XP_002055293, XP_002055407, XP_002055451, XP_002056569, XP_002056572, XP_002056889, XP_002057022, XP_002057023, XP_002057025, XP_002057046, XP_002057047, XP_002057312, XP_002057775, XP_002057863, XP_002058101, XP_002058167, XP_002058764, XP_002059026, XP_002059558, XP_002059924, XP_002062810, XP_002065107, XP_003249498, XP_003250408, XP_003436731, XP_004518508, XP_004519116, XP_004519322, XP_004520566, XP_004521010, XP_004523446, XP_004523968, XP_004524766, XP_004526337, XP_004529383, XP_004529412, XP_004530158, XP_004531038, XP_004531266, XP_004533385, XP_004533719, XP_004535304, XP_004536134, XP_004536932, XP_004537692, XP_005187487, XP_005190203,

XP_005190609, XP_006557645, XP_006558427, XP_006560286, XP_006563718, XP_006564634, XP_006564913, XP_006565977, XP_008191172, XP_008194238, XP_008199565, XP_008201416, XP_008201713, XP_019891905, XP_020713613, XP_021693932, XP_310951, XP_311387, XP_316164, XP_391836, XP_391836, XP_392478, XP_392899.2, XP_395280, XP_562185, XP_002056217, XP_966771, XP_967309,XP_967987, XP_968248,XP_969151, XP_969226, XP_969619, XP_970645, XP_971201, XP_971630, XP_972413, XP_972464, XP_973533, XP_975253, XP_EFA08711, AAEL002886, AAEL007001, AAEL007235, AAEL007392, AAEL008128, AAEL009387, AAEL010777, AAEL013980, AEGA01006707, AEGA01013763, AEGA01014751, AEGA01022594, AEGA01022595, AEGA01030422, AGAP000565RA, AGAP002277, AGAP004002, AGAP004002, AGAP004657, AGAP004657, AGAP007871, AGAP009584, AGAP009833, AGAP011107, AGAP012339, AHB50501, AAF46272, ALHF_02218.g1275, ALHF_11471, AQPM01000161, CH477216, CL1299, CL1818, CPIJ002542, CPIJ007967, CPIJ008049, CPIJ008889, EAA01572, EAT40089, EAT42717, EAT42717, EW987750, EX212033, FK813889, AEGA01003790, AEGA01018086, gi_145648988, gi_158703262, gi_78216392, gi309241287, GL501425, GL501437, GL630235, JP550838, NP_001014994, NP_001171496

- Official Gene Set (OGS): CG7975, CG10120, CG10748, CG10749, CG11401, CG1158, CG11611, CG12101, CG12157, CG13473, CG14690, CG15257, CG16954, CG17137, CG1724, CG18340, CG2137, CG2151, CG2830, CG3001, CG3057, CG31884, CG3215, CG32849, CG33102, CG3315, CG34132, CG3476, CG40451, CG4193, CG42302, CG43343, CG5495, CG5889, CG6492, CG6629, CG6647, CG6666, CG6852, CG7235, CG7311, CG7654, CG7964, CG7969, CG7998, CG8094, CG8256, CG8330, CG9042, CG9064

- Whole genome shotgun contigs (wgs), and Transcriptome Shotgun Assemblies TSA: GAWI01006724, GAWI01008942, GAMC01007766, GAWI01002350, GAWI01002349, GAWI01003186, GAWI01003819, GAWI01003871, GAWI01005176, GAWI01005478, GAWI01005502, GAWI01005514, GAWI01005625, GAWI01005678, GAWI01005947, GAWI01006090, GAWI01006179, GAWI01006311, GAWI01007175, GAWI01007422, GAWI01020148, GAWI01026563, GAMC01000264, GAMC01005605, GAMC01009657, GAMC01011211, GAMC01011214, GBBP01071272, GBBP01072246, GBBP01083257, GBBP01087487, GBBP01052257, GBBP01035041, GBBP01013511, GBBP01076789, GBBP01037147, GBBP01037184, GBBP01041269, GBBP01130520, GBBP01052076, GBBP01042577, GBBP01054104, GBBP01054167, GBBP01064369, GBBP01081694, GBBP01043905, GBBP01015449, GBBP01064676, GBBP01072051, GBBP01087855, GBBP01044033, GBBP01074354, GBBP01077960, GBBP01079055, GBBP01080199, GBBP01113631, GBBP01155502, GBBP01194611, GBBP01064676

- The National Center of Biotechnology Information (NCBI), and VectorBase: GMOY002421, GMOY002543, GMOY002684, GMOY003090, GMOY003575, GMOY004864, GMOY006103, GMOY006640, GMOY006832, GMOY007230, GMOY007568, GMOY007874, GMOY008210, GMOY009558, GMOY009911, GMOY010870, GMOY010871, GMOY011241, GMOY011667, GMOY011672, GMOY011701, GMOY012330, GMOY013299, GMOY013300, GMOY013302,

GMOY013303, LLOJ000608, LLOJ002548, LLOJ005819, LLOJ006374, LLOJ006852, LLOJ008094, LLOJ010089, LLOJ010089, MDOA000488, MDOA000548PA, MDOA001306, MDOA002306, MDOA002808, MDOA004695, MDOA006332PA, MDOA007103, MDOA007155PA, MDOA007400, MDOA007993, MDOA008701, MDOA009054, MDOA011431, MDOA011608, MDOA011810, MDOA012163, MDOA012708, MDOA013911, MDOA014337, MDOA014768, PPAIP000122, PPAIP000132, PPAIP003111, PPAIP003142PA, PPAIP003321, PPAIP006241, PPAIP006242, PPAIP010765.

## Supplemental Information

Supplemental information for this article can be found online at http://dx.doi.org/10.7717/peerj.10012#supplemental-information.

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
