# Peer review of "Genomic signatures of globally enhanced gene duplicate accumulation in the megadiverse higher Diptera fueling intralocus sexual conflict resolution"

_PeerJ, doi:10.7717/peerj.10012_

## Round 0.1 · original submission · Major Revisions

Dear Dr. Friedrich,

Your manuscript has been evaluated by two reviewers who are experts in the field.

Please follow and answer the comments of Reviewer #1 and Reviewer #2. Please try to address the comments of Reviewer #1 particularly regarding clarification of narrative regarding gene duplications in the Diptera lineage. Please also check for typos along the manuscript.
Reviewer #2 raised issues regarding appropriate literature citations.
I believe both reviewers will help you to improve the manuscript in a new version to PeerJ.

Reviewer 1 ·

Basic reporting

See general comments for the authors.

Experimental design

See general comments for the authors.

Validity of the findings

See general comments for the authors.

Additional comments

This study examines the occurrence of duplicated genes in the genomes of four holometabolous insects, including Drosophila melanogaster, Anopheles gambiae, Tribolium castaneum and Apis melifera. Highly duplicated gene families (above 6 paralogs) have been excluded. 698 gene families with lineage-specific duplications in Drosophila are reported compared 315 in Anopheles, 386 in Tribolium, and 223 in Apis. The Drosophila duplications are further explored by examining their synonymous substitution rates, GO terms, conservation in a brachyceran outgroup (hessian fly) and in various schizophoran species. Based on these analyses it is concluded that Brachycera experienced an accelerated retention rate of duplicated genes. Moreover, based on GO term analyses the authors suggest a significant enrichment of gene duplications related to energy metabolism. All of 17 gene families related energy metabolism found in the D. melanogaster lineage included previously known diverged germ-line specific paralogs, which leads the authors to conclude that the increased retention of gene duplications in the brachyceran lineage is not a result evolution of fast flight. Instead, they propose that germline subfunctionalization within these gene families could have resulted in intralocus sexual conflict resolution, thereby promoting clade size.

Critique
The finding that retention of gene duplicates is significantly higher in the D. melanogaster lineage (Schizophora lineage?) than in the mosquito, beetle or wasp lineage is interesting but there are a few issues in the author’s narrative that need clarification.

1. I do not understand how the authors can make any generalizations for the Brachycera clade without analyzing any basal-branching brachyceran lineages. For example, the data presented in this study are compatible with an acceleration of gene duplicate retention in Schizophora, Cyclorrhapha, or Brachycera. Cyclorrhapha are well supported and many changes have occurred in their stem lineage. This also applies to Schizophora, which underwent an accelerated radiation. This makes any discussion of what may have driven this evolutionary pattern more difficult. In my opinion this is an important limitation of the study. It would be appropriate to acknowledge and explain this limitation and to tune down speculations, in particular (but not only) at the end of the abstract.
2. The authors seem to rule out that the gene duplications affecting energy metabolism could have evolved in the context of fast flight. That might be the case but I do not follow the logic of their argument. Can’t this hypothesis be reconciled with an “out of the testis” framework for the origin of new genes, which by the way may not only apply to the 17 gene families with a link energy metabolism? A better test of the fast-flight-hypothesis would have been to examine gene duplication in basal-branching Brachycera.

Some minor points:
1. In the method section, I found it difficult to understand how gene duplications have been encoded. The authors could improve this section, e.g. by assigning in the specific examples each number in the code (1:0 etc) a specific meaning.
2. Line 72-75: the authors should also consider Lan & Prichard (Science 352, 1009-1013).
3. There are several typos in the manuscript, e.g. line 88, 141 (italics), 175, 188.
4. The reference to He (line 56) seems inappropriate. Why is the directive axis of Nematostella treated as a longitudinal axis? Does this study provide evidence for the orthologly of the Hox genes examined in Nematostella to those in bilaterians?

Reviewer 2 ·

Basic reporting

In this manuscript, Bao and Friedrich expanded the work in which they previously showed that Drosophila and other species from Brachycera group show an increased number of gene duplicates. To assess this, the authors focussed on duplications with less than 6 paralogs and compared with well annotated holometabolous species: mosquito, bee and red flour beetle.
From the gene families selected (with representatives in at least 3 of the 4 species), D. melanogaster has almost 700 vs a range between 386 (beetle)-221 (bee) in the other 3 species. They also investigated the evolution of these paralogs (synonymous substitutions) as a proxy for the age of expansions and found that more than 90% of duplicates have an ancient origin.
The authors also looked at the biological function of the duplicates (GO term enrichment). Drosophila duplicated gene families have GO categories related to Energy metabolism. Bao and Friedrich also checked other dipterans outside or inside the brachycerans to see whether those duplications were Brachycera specific or Diptera specific. All duplications preceded drosophilids and four gene family expansions were very ancient. They also looked gene expression using the ModENCODE database. The energy metabolism related gene families included germline-specific paralogs, which is an indicative of intralocus sexual conflict resolution. As examples, they showed the case of Hsp60 with subfunctionalization plus high divergence in the germline paralog and Hex gene family which does not fit into the intralocus sexual conflict resolution hypothesis.
The study is an expansion of previous work from the authors that will be relevant for the fields of evolutionary biology, evolutionary genomics and researchers working in the evolution of insects. The experiments are appropriate and properly performed and the results are clearly written along the manuscript. Therefore, I recommend the publication of this manuscript in PeerJ. However, I have some comments that should be addressed prior publication:

1. I am missing any mention (in the introduction and discussion) to the most recent literature regarding gene content and genomics in insects, especially the Genome Biology paper “Gene content evolution in the arthropods” and since one of the authors in this work is also a co-author in the GB paper. The results showed in this GB article are extremely relevant for this manuscript, thus, they should be mentioned and discussed.

2. On the other hand, there are several references to Li et al. 2018, which, in my opinion, concluded many WGD in Hexapoda based on very weak analyses (Genes ages inferred by Ks, age distribution of duplicates, mostly from transcriptomes data, no syntenic information…), thus, I recommend to remove this citation, or at least, to include also the response from Nakatani and McLysaght (PNAS, 2019) questioning the validity of that article.

3. lines 72- 75. Authors could comment on the recent findings from Marletaz et al. 2018 about the relevance of specialization of paralogs after WGD in vertebrates

Experimental design

The experiments are appropriate and properly performed

Validity of the findings

no comments

---

## Round 0.2 · Minor Revisions

Dear Dr. Friedrich,

I would like to congratulate and thank you for performing all the required changes in the manuscript.

Before the paper can be accepted I ask you to carefully revise the manuscript for typos throughout the text and some errors in scientific species names (which should be in italics).

Thank you and congratulations, it is a very interesting manuscript.

Reviewer 2 ·

Basic reporting

The authors have addressed the comments I had in the previous round of revision. Thus, I recommend its publication in PeerJ.
However, there are still some typos throughout the text and some errors in scientific species names (which should be in italics), therefore the manuscript should be carefully revised

Experimental design

no comment

Validity of the findings

no comment

Additional comments

The authors have addressed the comments I had in the previous round of revision. Thus, I recommend its publication in PeerJ.
However, there are still some typos throughout the text and some errors in scientific species names (which should be in italics), therefore the manuscript should be carefully revised in that matter

---

## Round 0.3 · accepted · Accept

Dear Dr, Friedrich,

Congratulations.

The manuscript has been accepted. Thanks a lot for submitting it to PeerJ, it is a great paper from your group.

Best regards
Rodrigo